



# Lyapunov analysis of multiscale dynamics:
# The slow manifold of the two-scale Lorenz '96 model

Mallory Carlu [1], Francesco Ginelli [1], Valerio Lucarini [2,3,4], and Antonio Politi [1]

[1]SUPA,Institute for Complex Systems and Mathematical Biology, King's College, University of Aberdeen, Aberdeen UK
[2]Department of Mathematics and Statistics, University of Reading, Reading, UK
[3]Centre for the Mathematics of Planet Earth, University of Reading, Reading, UK
[4]CEN, University of Hamburg, Hamburg, Germany

**Correspondence:** Mallory Carlu (mallory.carlu@abdn.ac.uk)

**Abstract.** We investigate the geometrical structure of instabilities in the two-scales Lorenz '96 model through the prism of Lyapunov analysis. Our detailed study of the full spectrum of covariant Lyapunov vectors reveals the presence of a *slow manifold* in tangent space, composed by a set of vectors with a significant projection on the slow degrees of freedom; they correspond to the smallest (in absolute sense) Lyapunov exponents and thereby to the longer time scales. We show that the

dimension of this manifold is extensive in the number of both slow and fast degrees of freedom, and discuss its relationship with the results of a finite-size analysis of instabilities, supporting the conjecture that the slow-variable behavior is effectively determined by a non-trivial subset of degrees of freedom. More precisely, we show that the slow manifold corresponds to the Lyapunov spectrum region where fast and slow instability rates overlap, "mixing"their evolution into a set of vectors which simultaneously carry information on both scales. We suggest these results may pave the way for future applications to ensemble

forecasting and data assimilations in weather and climate models.

## 1 Introduction

Understanding the dynamics of multiscale systems is one of the great challenges in contemporary science both in terms of theoretical aspects and applications in many areas of interests for the society and the private sectors. Such systems are characterized by having relevant dynamics taking place on diverse spatial and/or temporal scales, with interactions between different

scales combining with the presence of nonlinear processes. The presence of a variety of scales makes it hard to approach such systems using direct numerical integrations, since the problem immediately results as stiff. Additionally, usual arguments based on scale analysis, where only a limited set of scales are deemed important, usually fail because of the presence of, possibly slow, upward or downward cascade of energy and information.

     A crucial contribution for the understanding of multiscale systems comes from the now classic Mori-Zwanzig theory

(Zwanzig, 1960, 1961; Mori et al., 1974), which allows one to construct an effective dynamics specialised for the scale of interest, which are, typically, the slow ones. The enthusiasm one may have for the Mori-Zwanzig formalism is partly counterbalanced by the fact that the effective *coarse-grained* dynamics is written in an implicit form, so that it is of limited direct use. More tractable results can be obtained in the limit of an infinite time scale separation between the slow modes of interest and



the very fast degrees of freedom one wants to neglect: in this case, the homogeneisation theory indicates that the effect of the fast degrees of freedom can be written as the sum of a deterministic, drift-like correction plus a white noise stochastic forcing (Pavlioti et al., 2008).

The climate provides an excellent example of multiscale system, with dynamical processes taking place on a very large range
of spatial and temporal scales. The chaotic, forced, dissipative dynamics and the non-trivial interactions between different scales represent a fundamental challenge in predicting and understanding weather and climate. A fundamental difficulty in the study of the multiscale nature of the climate system comes from the lack of any spectral gap, namely, a clear and well-defined separation of scales. The climatic variability covers a continuum in terms of frequency (Peixoto and Oort, 1992; Lucarini et al., 2014), so that the powerful techniques based on homogeneization theory cannot be readily applied.

On the other side, there is a fundamental need to construct efficient and accurate parametrizations for describing the impact of small on larger scales in order to improve our ability to predict weather and provide a better representation of climate dynamics. Since some time it has been advocated that such parametrizations should include stochastic terms (Palmer and Williams, 2008). Such a point of view is becoming more and more popular in weather and climate modelling, even if the construction of parametrizations is mostly based on ad-hoc, empirical methods (Franzke et al., 2015; Berner et al., 2017). Weather and climate
applications have been instrumental in stimulating the derivation of new general results for the construction of paramerizations of multiscale systems, and for understanding the scale-scale interactions. Recent advances have been obtained using: (i) Mori-Zwanzig and Ruelle response theory (Wouters and Lucarini, 2012, 2013);(ii) the generalization of the homogenization theory-based results obtained via the Edgeworth expansion (Wouters et al., 2017); (iii) use of hidden Markov layers (Chekroun et al., 2015a, b) (from a data-driven point of view). An extremely relevant possible advantage of using theory-based methods is the
possibility of constructing scale-adaptive parametrizations; see discussion in (Vissio and Lucarini, 2017).

Another angle on multiscale systems deals with the study of the scale-scale interactions, which are key to understand instabilities and dissipative processes and the associated predictability and error dynamics. Lyapunov exponents (Pikovsky and Politi, 2016), which describe the linearized evolution of infinitesimal perturbations, are mathematically well established quantities, and seem to be the most natural choice to start addressing this problem. However, as it is well known, in multiscale systems the
maximum (or leading) Lyapunov exponent controls only the early-stage dynamics of very small perturbations (Lorenz, 1996). As time goes on, the amplitude of the perturbations of the fastest variables start saturating, while those affecting the slowest degrees of freedom grow at a pace mostly controlled by the (typically weaker) instabilities characteristic of the slower degrees of freedom. While fully nonlinear tools, such as finite-size Lyapunov exponents (Aurell et al., 1997), are able to capture the rate of this multi-scale growth, they lack the mathematical rigour of infinitesimal analysis. In particular, they are unable to convey
information on the leading directions of these perturbations, when they grow across multiple scales, an essential problem if one wishes to investigate at a deterministic level the non-trivial correlations across structures and perturbations acting on different scales. It is therefore of primary importance to better understand the multiscale and interactive structure of these instabilities and, in particular, to probe the sensitivity of multiscale systems to infinitesimal perturbations acting at different spatial and temporal scales and different directions. To this purpose, infinitesimal Lyapunov analysis allows one to compute not only a
full spectrum of Lyapunov Exponents (LEs), but also their corresponding *tangent space* directions, the so called covariant





Lyapunov vectors (CLVs) (Ginelli et al., 2007). CLVs are associated with LEs (in a relationship that, loosely speaking, resembles the eigenvector-eigenvalue pairing) and provide an intrinsic decomposition of tangent space that links growth (or decay) rates of (small) perturbations to physically based directions in configuration space. In principle, they can be used to associate instability time scales (the inverse of LEs) with well defined real-space perturbations or uncertainties.

While this information is gathered at the linearized level, one may nevertheless conjecture that LEs and CLVs associated with the slowest time scales (i.e. the smallest LEs in absolute value) can capture relevant information on the large-scale dynamics and its correlations with the faster degrees of freedom. In a sense, one may conjecture that the slow LEs and CLVs could be used to gain access to a non-trivial effective large scale dynamics. Accordingly, the identification of linear instabilities in full multiscale models, is then expected to have practical implications in terms of control and predictability.

In the following, we will begin to investigate these ideas studying the tangent-space structure of a simple two-scale atmospheric model, the celebrated Lorenz '96 (L96) model first introduced in (Lorenz, 1996).

The L96 model provides a simple yet prototypical representation of a two-scale system where large-scale, synoptic variables are coupled to small-scale, convective variables. The Lorenz 96 model has quickly established as an important testbed for evaluating new methods of data assimilation (Trevisan and Uboldi, 2004; Trevisan et al., 2010) and stochastic parametrizations

schemes (Vissio and Lucarini, 2017; Orrel, 2003; Wilks, 2006) In the latest decade, it also received considerable attention in the statistical physics community (Abramov and Majda, 2007; Hallerberg et al., 2010; Lucarini and Sarno, 2011; Gallavotti and Lucarini, 2014), while an earlier study – limited to the stronger instabilities – has highlighted the localization properties of the associated CLVs (Herrera et al., 2011).

We anticipate that our infinitesimal Lyapunov analysis reveals the existence of a nontrivial *slow manifold* in tangent space,

formed by a set of CLVs – associated with the smallest LEs – the only ones to have a non-negligible projection on the slow variables. The number of these CLVs is considerably larger than the mere number of slow variables, and it is *extensive* in the number of slow and fast degrees of freedom. On the other hand, directions associated with the highly expanding or contracting LEs are aligned almost exclusively along the fast, small-scale degrees of freedom. Also, we show that the LE corresponding to the first CLV of the slow manifold (i.e. the most expanding direction of such a manifold) approaches the large perturbation

limit of the finite size Lyapunov exponents. This result suggests that Lyapunov analysis is indeed able to capture an effective *deterministic* representation of the large scale dynamics which also involves relevant fast degrees of freedom in addition to the slow ones, representing the first step in the program sketched above.

The remaining of this paper is organized as follows. Section 2 introduces both the L96 model and the fundamental tools of Lyapunov analysis used in this paper. Our main results, the existence of a tangent-space slow manifold, is presented in section

3. In Section 4, we investigate how this manifold arises from the superposition of the instabilities of the slow and fast dynamics respectively. Section 5, on the other hand, is devoted to a comparison with results of finite-size analysis. Finally, in Section 6 we discuss our results, further commenting on their generality and proposing future developments and applications.





## 2 The Lorenz '96 model: a simple multiscale system

### 2.1 Model definition and scaling considerations

The L96 model is a simple example of an extended multi-scale system such as the Earth atmosphere. Its dynamics is controlled by synpoptic variables, characterized by a slow evolution over large scales, coupled to the so-called convective variables characterized by a faster dynamics over smaller scales.

The synoptic variables $X_k$, with $k = 1, ..., K$, represent generic observables on a given latitude circle; each $X_k$ is coupled to a subgroup of $J$ convective variables $Y_{k,j}$, $(j = 1, ..., J)$ that follow the faster convective dynamics typical of the $k$ sector,

$$\dot{X}_k = X_{k-1}(X_{k+1} - X_{k-2}) - X_k + F_s - \frac{hc}{b} \sum_j Y_{k,j} \tag{1a}$$

$$\dot{Y}_{k,j} = cbY_{k,j+1}(Y_{k,j-1} - Y_{k,j+2}) - cY_{k,j} + \frac{c}{b}F_f + \frac{hc}{b}X_k \tag{1b}$$

In both sets of equations, the nonlinear nearest-neighbor interaction provides an account of advection due to the movement of air masses, while the last terms describe the mutual coupling between the two sets of variables. Each $X_k$ variable is affected by the sum of the associated $Y_{k,j}$ variables, while each $Y_{k,j}$ is forced by the variable $X_k$ corresponding to the same sector $k$. Finally, the linear terms $-X_k$ and $-cY_{k,j}$ account for internal dissipative processes (viscosity) and are responsible for the contractions of the phase space.

We remark that in our configuration, following (Vissio and Lucarini, 2017), energy is injected in the system both at large and at small scales, provided by the constant terms $F_s$ and $F_f$, which impact the slow and fast scales of the system, respectively.

The presence of the additional forcing term acting on the $Y_{k,j}$ variables makes it possible to have a chaotic dynamics on the small scales also in the limit of vanishing coupling ($h \rightarrow 0$), as opposed to the typical L96 setting, where the small-scale variables become spontaenously chaotic without the need of being forced by their associated $X_k$ as a result of downward energy cascade from the slow variables.

Moreover, the parameter $c$ controls the time-scale separation between the $X_k$ and $Y_{k,j}$ variables, while $b$ controls their relative amplitude. Finally, $h$ gauges the strength of the coupling between slow and fast variables.

The L96 model thus contains $K$ slow variables and $K \times J$ fast variables, for a total of $N = K(1 + J)$ degrees of freedom. It is complemented by the boundary conditions

$$X_{k-K} = X_{k+K} = X_k$$

$$Y_{k-N,j} = Y_{k+K,j} = Y_{k,j} \tag{2}$$

$$Y_{k,j-J} = Y_{j,k-1}$$

$$Y_{k,j+J} = Y_{k+1,j}$$

In his original work (Lorenz, 1996), Lorenz considered $K = 36$ slow variables and $J = 10$ fast variables for each subsector, for a total of $N = 396$ degrees of freedom. As usual, one is ideally interested in dealing with arbitrarily large $K$ and $J$ values, so that it is preferable to formulate the model in such a way that it remains meaningful in the limit $K, J \rightarrow \infty$. In this respect,





the only potential problem is the global coupling, represented by the sum in Eq. (1a), which should stay finite for $J \to \infty$. This can be easily ensured by imposing that the coefficient in front of the sum is inversely proportional to $J$. The most compact representation is obtained by introducing the rescaled variables $Z_{k,j} = bY_{k,j}$, and replacing $b$ with a new parameter $f$

$$f = \frac{Jc}{b^2}, \tag{3}$$

With these transformations, Eqs. (1) can be rewritten as

$$\dot{X}_k = X_{k-1}(X_{k+1} - X_{k-2}) - X_k + F_s - hf\langle Z_{k,j}\rangle_j \tag{4a}$$

$$\frac{1}{c}\dot{Z}_{k,j} = Z_{k,j+1}(Z_{k,j-1} - Z_{k,j+2}) - Z_{k,j} + F_f + hX_k \tag{4b}$$

where

$$\langle Z_{k,j}\rangle_j = \frac{1}{J}\sum_{j=1}^{J} Z_{k,j}, \tag{5}$$

while the boundary conditions are the same as above. In practice $f$ gauges the asymmetry of the interaction between slow and fast variables. From its definition, it is clear that $f$ strongly depends on the scale separation $b$. For the standard choice of the parameter values (see below), $f = 1$, i.e. the average influence of the fast scales on the slow ones is the same as the opposite. On the other hand, if we increase the value of $b$, $f \to 0$, which corresponds to a master-slave limit, where the fast variables do not affect the slow ones but are actually slaved to them. This makes sense because the small-scale variables have extremely

small amplitude. The opposite master-slave limit, pherhaps more interesting from a climatological point of view, corresponds to taking the $h \to 0$ and $f \to \infty$ limits, while keeping the product $hf$ constant. In this case, the fast variables follow up to first approximation their own autonomous dynamics, but still drive the slow ones through the finite coupling term $hf\langle Z_{k,j}\rangle_j$. In this latter limit, we envision the presence of an upscale energy transfer.

      Apart from helping to clarify these master-slave limiting cases, such a reformulation of the model also allows also to better

understand that, in order to maintain a fixed amplitude of the coupling term, it is necessary to keep $f$ constant when $J$ is varied. Selecting a constant value for the time-scale separation $c$, we choose to rescale $b$ with $J$, as follows:

$$b = \sqrt{\frac{Jc}{f}}. \tag{6}$$

With reference to the Lorenz original parameter choices (Lorenz, 1996), $b = c = 10$ and $J = 10$, we have $f = 1$ and the suggested scaling

$$b = \sqrt{10J}. \tag{7}$$

It is finally interesting to note that, in the absence of forcing and dissipation, Eqs. (1) reduce to

$$\dot{X}_k = X_{k-1}(X_{k+1} - X_{k-2}) - \frac{hc}{b}\sum_j Y_{k,j} \tag{8a}$$

$$\dot{Y}_{k,j} = cbY_{k,j+1}(Y_{k,j-1} - Y_{k,j+2}) + \frac{hc}{b}X_k \tag{8b}$$





which conserve a quadratic form of slow and fast variables (Vissio and Lucarini, 2017)

$$E = \sum_k X_k^2 + \sum_{k,j} Y_{k,j}^2 = \sum_k \left( X_k^2 + \frac{f}{c} \left\langle Z_{k,j}^2 \right\rangle_j \right) \tag{9}$$

This conservation law, of course, does not hold in the more interesting forced and dissipative case. However, this result suggests that $E$ can be identified with a bona fide energy – and represents a natural norm – also in the forced and dissipative case.

Note also that, according to the last equality in Eq. (9), changing the number of fast variables does not change the total energy budget, provided that the ratio $f/c$ remains constant.

In this study, unless otherwise specified, we will set $f = 1$ and typically adopt the slow forcing and the time-scale separation originally adopted by Lorenz, $F_s = 10$ and $c = 10$, and choose values for $b$ and $J$ that satisfy the scaling condition (7). Ac-

cording to Lorenz's original derivation, one time unit in this model dynamics is roughly equivalent to 5 days in the real climate evolution (Lorenz, 1996).

We will fix $F_f = 6$, which guarantees chaoticity in the uncoupled fast variables in the absence of coupling. Lorenz's original choice for the coupling between the slow and fast scales was $h = 1$, but here we will also explore the weak coupling regime, considering coupling values as small as $h = 1/16$.

## 2.2 Elements of Lyapunov analysis: Lyapunov Exponents and Covariant Lyapunov Vectors

As mentioned above, the right tools to quantify rigorously the rate of divergence (or convergence) of nearby trajectories, are the Lyapunov characteristic exponents (LEs) and their associated covariant Lyapunov vectors (CLVs). We provide here a qualitative description of these objects. For a more thorough discussion, the reader can look at (Ruelle, 1979; Eckmann and Ruelle, 1985; Ginelli et al., 2013; Kuptsov and Parlitz, 2012) and references therein.

For definiteness, let us consider an $N$ dimensional continuous-time dynamical system

$$\dot{\mathbf{x}}(t) = \mathbf{f}(\mathbf{x}(t)), \tag{10}$$

with $\mathbf{x}(t)$ being the state of the system at time $t$. One can linearize the dynamics around a given trajectory, thus obtaining the evolution of an infinitesimal perturbation $\delta \mathbf{x}(t)$ in the so-called *tangent space*

$$\delta \dot{\mathbf{x}}(t) = \mathbf{J}(\mathbf{x}, t) \delta \mathbf{x}(t), \tag{11}$$

where we have introduced the *Jacobian* matrix

$$\mathbf{J}(\mathbf{x}, t) = \frac{\partial \mathbf{f}(\mathbf{x}(t))}{\partial \mathbf{x}(t)}. \tag{12}$$

LEs $\lambda_i$ measure the (asymptotic) exponential rates of growth (or decay) of infinitesimal perturbations along a given trajectory. Their plurality holds in the fact that the growth rates associated with different directions of the infinitesimal perturbations are in general different. We then refer to the ordered sequence $\lambda_1 \geq \lambda_2 \geq ... \geq \lambda_N$ as to the spectrum of characteristic LEs, or





Lyapunov Spectrum (LS), with $N$ being the dimension of the dynamical system. At each point $\mathbf{x}(t)$ of the attractor, the CLVs $\mathbf{v}_i(\mathbf{x}(t))$ give the directions of growth of perturbations associated with the corresponding Lyapunov exponent[1]. In other words, they span the Oseledets splitting (Eckmann and Ruelle, 1985), i.e. an infinitesimal perturbation $\delta\mathbf{x}_i(t_0)$ exactly aligned with the $i$th CLV $\mathbf{v}_i(\mathbf{x}(t_0))$, after a sufficiently long time $t$ will grow or decay as

$$\|\delta\mathbf{x}_i(t_0+t)\| \approx \|\delta\mathbf{x}_i(t_0)\| e^{\lambda_i t}. \tag{13}$$

LEs are global quantities, measuring the average exponential growth rate along the attractor, while CLVs are local objects, defined at each point of the attractor and transforming covariantly along each trajectory, according to the linearized dynamics (11),

$$\mathbf{v}(\mathbf{x}(t)) = \mathbf{M}(\mathbf{x}_0,t)\mathbf{v}(\mathbf{x}_0), \tag{14}$$

where $\mathbf{x}_0 \equiv \mathbf{x}(0)$ and the *tangent linear propagator* $\mathbf{M}(\mathbf{x}_0,t)$ satisfies

$$\dot{\mathbf{M}}(\mathbf{x}_0,t) = \mathbf{J}(\mathbf{x},t)\mathbf{M}(\mathbf{x}_0,t), \tag{15}$$

with $\mathbf{M}(\mathbf{x}_0,0)$ being the identity matrix.

CLVs constitute an intrinsic (like LEs they do not depend on the chosen norm) tangent-space decomposition into the stable and unstable directions associated with the different LEs. LEs themselves have units of inverse time, so that the largest positive (in absolute value) exponents – and their associated CLVs – describe fast growing (or contracting) perturbations, while the smaller ones corresponds to longer time scales.

Unfortunately, Eqs. (13-14) cannot be used to compute directly any LEs or CLVs beyond the first one. Unavoidable numerical errors generated while handling higher-order CLVs are amplified according to a rate dictated by the largest LE, so that any tangent-space vector quickly converges to the first CLV. In order to avoid this collapse, it is customary to periodically orthonormalize the vectors with a QR-decomposition (Shimada and Nagashima, 1979; Benettin et al.., 1980). LEs are thereby computed as the logarithms of the basis vectors normalization factors, time-averaged along the entire trajectory.

The mutually orthogonal vectors, obtained as a by-product of this procedure, constitute a basis in tangent space and are usually referred to as Gram-Schmidt vectors (by the name of the algorithm used to perform the QR-decomposition) or Backward Lyapunov Vectors (BLVs, because they are obtained by forward integrating the system until a given point in time, thus spanning the past trajectory with respect to this point). Being forced to be mutually orthogonal, BLVs allow only reconstructing the orientation of the subspaces spanned by the most expanding directions. In this work, we concentrate on the CLVs for the identification of the various expanding/contracting direction. This is done by implementing a dynamical algorithm, based on a clever combination of both forward and backward iterations of the tangent dynamics, introduced in (Ginelli et al., 2007) and more extensively discussed in Ref. (Ginelli et al., 2013).

In practice, one first evolves the forward dynamics, following a phase space trajectory to compute the full LS $\{\lambda_i\}_{i=1,...,N}$ and

---

[1]In the presence of $m > 1$ degenerate (i.e. identical) LEs, the corresponding $m$ CLVs span an $m$-dimensional Oseledets subspace whose elements are all characterized by the same growth rate.





the basis of BLVs $\{\mathbf{g}_i(t_m)\}_{i=1,\ldots,N}$ with a series of QR-decompositions performed along the trajectory every $\tau$ time units, at times $t_m = m\tau$, with $m = 1,\ldots,M$. One is then left with a series of orthogonal matrices $\mathbf{Q}_m$, whose columns are the BLVs $\mathbf{g}_i(t_m)$ and the upper triangular matrices $\mathbf{R}_m$ which contain the vectors norms and their mutual projections.

The key idea is then to project a generic tangent space vector $\mathbf{u}(t_m)$ on the covariant subspaces $S_j(t_m)$ spanned by the first $j$

BLVs at times $t_m$. It can be easily shown (Ginelli et al., 2013) that this projection, evolved backward in time according to the inverse tangent-space dynamics, converges exponentially fast to the $j$th covariant vector.[2] In practice, this backward procedure can be performed by expressing the CLVs in the BLVs basis,

$$\mathbf{v}_j(t_m) = \sum_{i=1}^{j} c_{i,j}(t_m)\mathbf{g}_i(t_m). \tag{16}$$

The coefficents $c_{i,j}(t_m)$ thus compose an upper triangular matrix $\mathbf{C}_m$, whose dynamics is actually determined by the $\mathbf{R}_m$

matrices obtained from the QR-decomposition

$$\mathbf{C}_m = \mathbf{R}_m \mathbf{C}_{m-1}. \tag{17}$$

This last relationship is very easily invertible, assuring a computationally efficient and precise method to follow the backwards dynamics.

### 2.3 Lorenz '96 tangent-space dynamics and algorithmic aspects

The tangent space dynamics of L96 can be readily obtained by linearizing the phase space evolution equation (1),

$$\delta\dot{X}_k = \delta X_{k-1}\left(X_{k+1} - X_{k-2}\right) + X_{k-1}\left(\delta X_{k+1} - \delta X_{k-2}\right) - \delta X_k - \frac{hc}{b}\sum_j \delta Y_{k,j} \tag{18a}$$

$$\delta\dot{Y}_{k,j} = cb\big[\delta Y_{k,j+1}\left(Y_{k,j-1} - Y_{k,j+2}\right) + Y_{k,j+1}\left(\delta Y_{k,j-1} - \delta Y_{k,j+2}\right)\big] - c\delta Y_{k,j} + \frac{hc}{b}\sum_j \delta X_k\,, \tag{18b}$$

where $\delta X_k$ and $\delta Y_{k,j}$ are infinitesimal perturbations of, respectively, slow and fast variables. Together, they define the tangent space vector $\mathbf{u} \equiv (\delta X_1,\ldots,\delta X_K, \delta Y_{1,1},\ldots \delta Y_{K,J})$. One can easily deduce the Jacobian matrix from Eqs. (18).

In this paper, we numerically integrate Eqs. (1,18) using a Runge-Kutta 4th order algorithm, with a time step $\Delta t = 10^{-3}$, shorter that the choice $\Delta t = 5 \cdot 10^{-3}$ typically made for the standard L96 model. In fact, we have verified that such a short time step is actually required in order to compute the entire spectrum of LEs and CLVs with a sufficient accuracy. Typically, to discount transient effects in numerical simulations, we discard the first $10^3$ time units, split in two equal parts: the first 500 time-units allow for the phase space trajectory to reach its attractor, while the second is used for the convergence of the tangent-

space vectors towards the BLVs basis. Afterwards, a forward integration of typically $T = 10^3$ time units is performed in order to analyze the properties of tangent space. Due to the highly unstable nature of the L96 model (we will see in the following that the maximum LE is around 20, for our choice of parameter values), we have to perform the tangent space orthonormalization every $\tau = 10^{-2}$ time units Finally, a transient of $10^2$ time units is used during the backward dynamics to ensure the convergence

---

[2]Or, in the case of degenerate LEs, to a vector belonging to the corresponding Oseledets subspace.

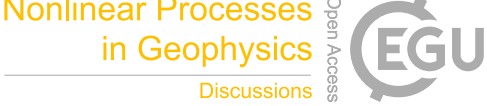



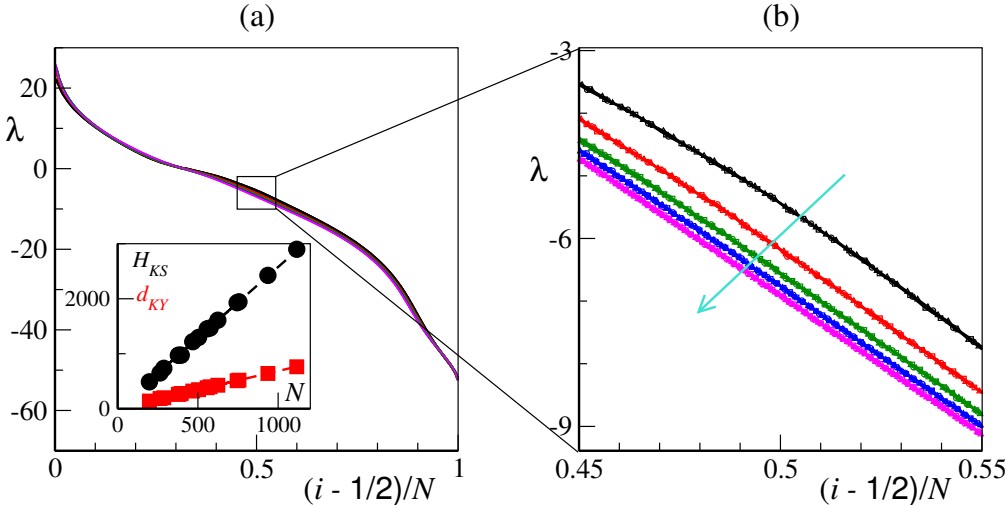

**Figure 1.** Extensivity of chaos. (a) Lyapunov spectra as functions of the rescaled index $i_r = (i - 1/2)/N$ for $K = 18, 24, 36$ and $J = 10, 15, 20, 25, 30$ (all possible combinations). (b) Detail of the central region $i_r \in (0.45, 0.55)$. The cyan arrow marks the direction of increasing $J$ values, while each $J$-branch is the superposition of the spectra for $K = 18$ (circles), $K = 24$ (squares) and $K = 36$ (triangles). Inset of panel (a): Kolmogorov-Sinai entropy $H_{KS}$ (black circles) and Kaplan-Yorke dimension $d_{KY}$ (red squares) as a function of the number of degrees of freedom $N = K(J + 1)$. The dashed lines mark a linear fit with zero intercept and slope $\approx 2.6$ ($H_{KS}$) and $\approx 0.7$ ($d_{KY}$). Simulations have been performed with $h = 1$ and $b = \sqrt{10J}$ (see main text).

of the backwards vectors to the true CLVs. We have also carefully verified that the forward and backward transients are long enough to guarantee a sufficiently accurate convergence to the true LEs and CLVs.

## 2.4 The Lorenz '96 Lyapunov spectrum

Spatially extended systems are known to typically exhibit an extensive Lyapunov spectrum (Ruelle, 1978; Livi et al., 1986; Grassberger, 1989). This property is instrumental for the identification of intensive and extensive observables in the thermodynamic sense. The single-scale L96 model (i.e. Eq. (1a) without the coupling to the fast scale) is no exception (Karimi and Paul, 2010), and here we show that the extensivity of chaotic behavior also holds in the two-scale model provided that – as discussed in Sec. 2.1 – the scaling relation (6) is fulfilled. According to our parameter choice, we set $b = \sqrt{10J}$ and proceed to systematically vary both $K$ and $J$. In the following we illustrate the case $h = 1$, but analogous results hold for different coupling parameters. Lyapunov spectra, plotted in Fig. 1a as functions of the rescaled index $(i - 1/2)/N$ for different numbers $K, J$ of slow and fast variables per subsector, show indeed a clear collapse. As a consequence, the Kolmogorov-Sinai entropy $H_{KS}$, a measure of the diversity of the trajectories generated by the dynamical system, is expected to be proportional to the number $N$ of degrees of freedom. This can be appreciated in the inset of Fig. 1a (see the black circles), where $H_{KS}$ is determined through



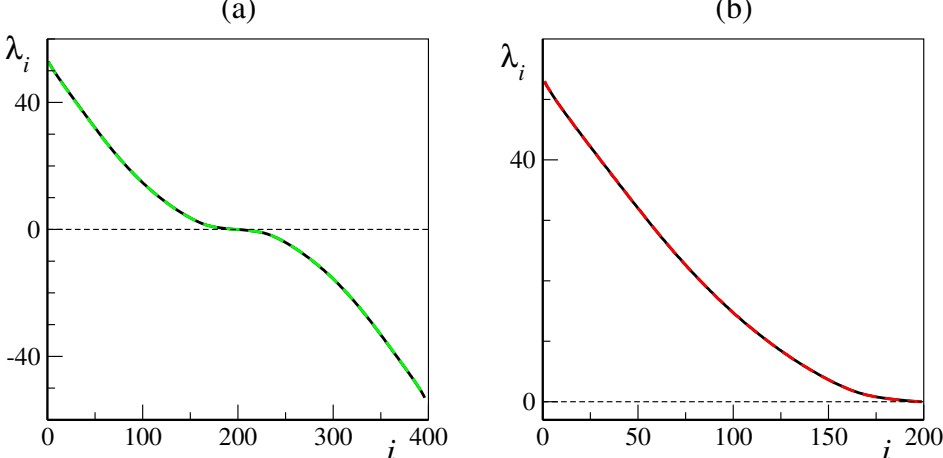

**Figure 2.** (a) Lyapunov Spectrum for the conservative setup, Eq. (8). Parameters are $b = 10$, $K = 36$, $J = 10$ ($N = 396$ LEs), and initial conditions are chosen such that the conserved energy is $E = 36$. The full black line corresponds to $h = 1$, while the dashed green line corresponds to $h = -1$. (b) The second half of the Lyapunov spectrum ($i \in [199, 396]$, dashed red line) is folded under the reflection transformation $\lambda_i \rightarrow -\lambda_{N-i+1}$ over its first half ($i \in [1, 198]$, full black line).

the Pesin formula (Eckmann and Ruelle, 1985)

$$H_{KS} = \sum_{\lambda > 0} \lambda_i. \tag{19}$$

Similarly, the dimension of the attractor, i.e. the number of active degrees of freedom, is proportional to $N$, as seen again from Fig. 1a (see the red squares), where the dimension $D_{KY}$ is determined from the Kaplan-Yorke formula (Eckmann and Ruelle, 1985)

$$D_{KY} = M + \frac{\sum_{i \leq M} \lambda_i}{|\lambda_{M+1}|}, \tag{20}$$

with $M$ being the largest integer such that $\sum_i \lambda_i > 0$. The distinct role of the two parameters $K$ and $J$ in the convergence to the thermodynamic limit can be better appreciated in Fig. 1b, which zooms on the central part of the spectrum. There, the single spectra are clearly grouped in five different branches, each corresponding to three different values of $K$ ($K = 18, 24, 36$, respectively marked as circles, squares and triangles) and to the same number $J$ of fast variables. As $J$ is increased from 10 to 30, these branches clearly converge to a limiting, thermodynamic spectrum. Note however that the convergence inside each branch, which corresponds to increasing $K$ while keeping $J$ fixed, is practically perfect up to our numerical precision. In a sense, convergence to the asymptotic Lyapunov spectrum is much faster in $K$ than in $J$: this is not a surprise, since the number of fast variables is $KJ$.

We conclude this section with a brief remark on the Lyapunov spectrum in the zero-dissipation limit (8), shown in Fig. 2a for $h = 1$. The LS is independent of the initial condition, provided that it lies on the same energy hypersurface (in our simulations


we have chosen $E = 36$ – see Eq. (9))[3]. Moreover, from Fig. 2b, we can appreciate that the LS is perfectly symmetric, since the second half of the spectrum superposes to the first half under the transformation $\lambda_i \to -\lambda_{N-i+1}$. This symmetry is an unexpected general property, which holds for any choice of $h$, $c$ and $b$. In fact, the existence of a conservation law can be invoked only for the existence of an extra zero-Lyapunov exponent. The overall symmetry of the LS must follow from more general properties such as the symplectic structure of the model, or invariance under time-reversal of the evolution equations.

5   Unfortunately, this model is known to possess no symplectic structure, even if the energy is conserved (Blender et al., 2013) and the only symmetry we have been able to find is the invariance under the transformation $t \to -t$, $X_k \to -X_k$, $Y_{k,j} \to -Y_{k,j}$, accompanied by a change of the coupling constant $h$. Indeed, the dashed green line in Fig. 2a shows that the Lyapunov spectrum is invariant under the transformation $h \to -h$. Therefore, the overall symmetry remains an unexplained property.

## 3   Slow tangent-space manifold

### 3.1   Projection of CLVs in the X subspace

We now come to the central result of this paper, namely the existence of a nontrivial manifold in tangent space associated with the slow dynamics of the L96 model.

The individual LEs $\lambda_i$ represent the average growth rate (and thus the inverse of suitable time scales) of well defined small perturbations aligned along the corresponding CLV, $\mathbf{v}_i$. It is therefore logical to ask which of these "fundamental" perturbations are more relevant for the evolution of the accessible macroscopic observables. In the present case, it is natural to focus our attention on the alignment along the slow variables $X_k$.

The norm of the (rescaled) $i$th CLV can be written as

$$||\delta X^{(i)}||^2 + ||\delta Y^{(i)}||^2 = 1 \,, \tag{21}$$

20   where the two addenda represent the squared Euclidean norm of the projection onto the slow and fast variables, $\delta X^{(i)} = (\delta X_1^{(i)}, \dots, \delta X_K^{(i)})$ and $\delta Y^{(i)} = (\delta Y_{1,1}^{(i)}, \dots \delta Y_{K,J}^{(i)})$ respectively. The most natural indicator of how much the $i$th CLV projects on the slow modes is thus the $X$-projected norm $\phi_i \equiv ||\delta X^{(i)}||^2$.

Given the strong temporal fluctuations of $\phi_i(t)$ when the vectors are covariantly transformed along a trajectory (see the end of this section), it is convenient to refer to its time average (which, assuming ergodicity, corresponds to an ensemble average over the invariant measure),

$$\Phi_i = \langle \phi_i(t) \rangle_t \,. \tag{22}$$

Before proceeding with our analysis, we wish to remark that, although the CLVs are intrinsic vectors, their projection $\phi_i$ does depend on the relative scales used to represent the fast and slow variables. If we change the scale of the fast $Y$ variables,

---

[3]This suggests that the energy shell is fully covered by the "attractor", i.e. that we have full measure and thereby ergodicity within the shell itself (Gallavotti and Lucarini, 2014).



introducing $V_j = \gamma Y_j$ the (Euclidean) norm of the $i$th CLV can be written as

$$L = \phi_i + \gamma^2 (1 - \phi_i).$$

After renormalizing all components to unit norm, it follows that the weight of the projection on the slow variables becomes

$$\phi'_i = \frac{\phi_i}{\phi_i(1 - \gamma^2) + \gamma^2} \,,$$

which shows how the amplitude of the projection depends on the relative scale between fast and slow variables. Since in the very definition of energy (see Eq. (9)), $X$ and $Y$ variables are weighted in the same way, it is natural to maintain the original definition, i.e. to assume $\gamma = 1$. Nevertheless, as we will see while discussing the evolution of finite perturbations, the relative scale is an important parameter we can play with to extract useful information.

We have first computed the projection norm $\Phi_i$ for the entire spectrum of vectors in a system of size $K = 36$ and $J = 10$ with the "standard" parameter value $b = 10$. In a wide range of coupling strengths – from strong ($h = 1$) to weak ($h = 1/16$) – we find that both rapidly growing and rapidly contracting perturbations are almost orthogonal to the slow-variable subspace, the associated CLVs exhibiting a negligible projections over the $X$ directions (see Fig. 3a). In fact, only a "central band" constituted by the CLVs associated with the smallest LEs display a significative projection over the slow variables.

Note also that the CLV associated with the null LE (in the following we simply denote it as the 0-CLV) displays a sharp peak of the projection norm $\Phi_i$. This is just a consequence of the well known delocalization of this CLV: the perturbation corresponding to the zero exponent points exactly along the trajectory, and thus by construction has relevant components in the slow-variable directions[4]. This central band of CLVs defines the tangent-space *slow manifold* relevant for this paper. It becomes more sharply defined for small values of the coupling $h$ but it keeps approximately the same position and width as

the coupling $h$ is increased. In particular, for this set of parameter values, this non-trivial *slow manifold* extends in tangent space over roughly 120 CLVs, much more than the $K = 36$ slow degrees of freedom. The extension of the slow manifold can be better appreciated in Fig. 3b, where the time-averaged projections are shown in logarithmic scale (top panel) and compared with the full spectrum of LEs (bottom panel). We are interested in the dependence of this manifold on the number of slow and fast variables. As discussed in the previous section, the L96 model is extensive in both the slow and fast variables, provided

that the ratio $f = Jc/b^2$ is kept constant (for the "standard" choice of parameters $c = 10$ and $f = 1$, so that it is sufficient to set $b = \sqrt{10J}$). In the following we present results for $h = 0.5$, but we have carefully verified that analogous results hold for other values of the coupling constant $h$.

We first set $J = 10$ and explore the behavior of the slow manifold when $K$ is varied (note that no parameter rescaling is required while changing $K$). Our simulations, reported in Fig. 4a, clearly show that the slow manifold is extensive w.r.t. $K$:

upon rescaling the vector index as $i \to (i - 0.5)/K$ we observe a clear collapse of the projection patterns[5].

We next focus on the scaling with $J$ at fixed $K$. For the sake of computational simplicity, we first consider $K = 18$. As for the

---

[4]Provided that, as we have verified, a notable part of the energy is located in the slow variables.

[5]The index $i = 1, \ldots, N$ is customarily shifted by $1/2$ units to ensure a faster convergence to the limit spectrum.





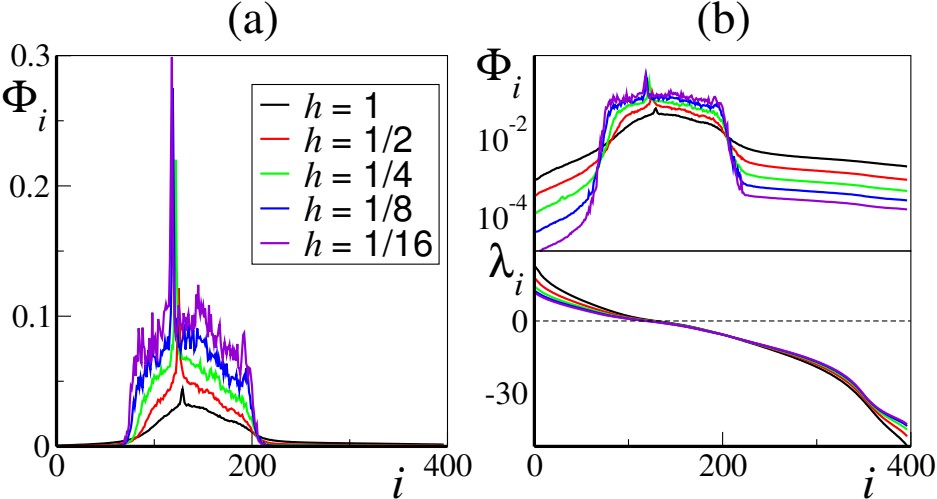

**Figure 3.** CLVs projection on the slow variables. (a) CLVs average projection norm $\Phi_i$ of the CLVs (see text) as a function of the vector index for $K = 36$, $J = 10$ and $b = 10$ upon varying the coupling constant $h$. (b) Upper panel: same as in (a) but in a logarithmic scale vertical scale. Bottom panel: Corresponding Lyapunov Spectra.

horizontal variable, it is natural to rescale the CLVs index by the number $J$ of fast degrees of freedom per subgroup. Moreover, since $J$ corresponds to the ratio between the number of fast ($KJ$) and slow ($K$) variables and we use the standard Euclidean norm to quantify the projection on the $X$ subspace, one expects the fraction $\Phi_i$ of $X$-norm to be inversely proportional to $J$. The projection data reported in Fig. 4b indeed show a convincing vertical collapse of the rescaled $X$-norm $\Phi_i J/J_0$ (here we

fix a reference $J_0 = 10$), but accompanied by a shrinking of the central band on the right side, as $J$ is increased.

In order to accurately determine the width of the central band, i.e. the slow-manifold dimension, we fix a threshold for the *rescaled* $X$-norm, $J\Phi_i/J_0 = 10^{-2}$, and estimate the number $N_s$ of CLVs with a projection above such a threshold (we have verified that our results hold within a reasonable range of thresholds). The resulting widths $N_s(K, J)$, computed for different

numbers of slow variables $K$, are summarized in Fig. 4c, where they are plotted versus $J$. For fixed $K$, we see a clear linear increase, compatible with the law

$$N_s(K, J) = K(1 + \alpha J) \tag{23}$$

where the coefficient $\alpha$ depends on the values of $h$, $c$, $F_s$ and $F_f$. In the present case, a best fit gives $\alpha \approx 0.22$. The most general representation of the projections is finally obtained by rescaling the index according to $N_s$ and by using the 0-CLVs (which

corresponds to the peak of $\Phi_i$) as the origin of the horizontal axis. The excellent collapse in Fig. 4d confirms the extensivity of the slow manifold with both $K$ and $J$. The slow manifold is not a simple represenation of the $X$ subspace: it does not concide with the slow variables themselves, but involves also a finite fraction $\alpha$ of the fast ones, singling out a fundamental set of tangent space perturbations closely associated with the slow dynamics. The origin of the phenomenological scaling law (23)





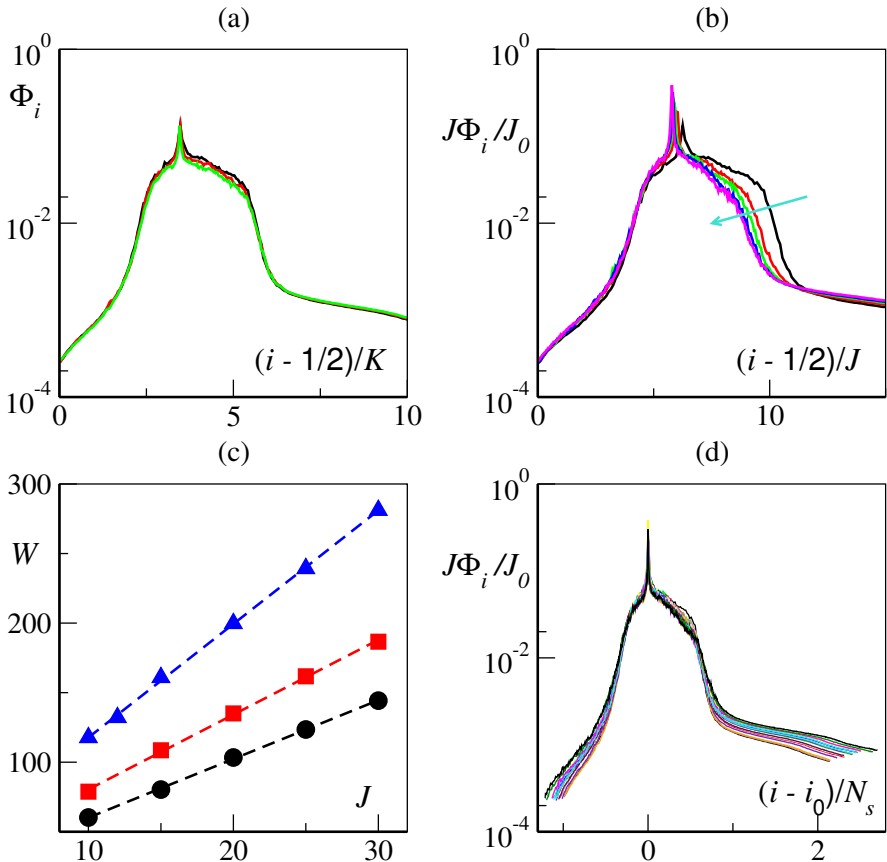

**Figure 4.** Slow manifold scaling for $h = 0.5$ and $f = 1$. (a) CLVs time-averaged X-projection norm $\Phi_i$ for $K = 18, 24, 36$ and $J = 10$ vs. the rescaled index $(i - 0.5)/K$. (b) Rescaled (see main text) CLVs average projection norm $\Phi_i$ for $K = 18$ and $J = 10, 15, 20, 25, 30$ (increasing along the cyan arrow) vs. the rescaled index $(i - 0.5)/J$. A lack of precise collapse can be appreciated on the rightmost side of the "central band". (c) Central band width $W$ (see main text for details) as a function of $J$ for $K = 18$ (black circles), $K = 24$ (red squares) and $K = 36$ (blue triangles). The best linear fits, marked by the dashed lines are $W = 18(1) + 4.20(5)J$ (for $K = 18$), $W = 26(2) + 5.4(1)J$ ($K = 24$) and $W = 36(2) + 8.2(2)J$ ($K = 36$). (d) Rescaled CLVs average projection norm $\Phi_i$ for $K = 18, 24, 36$ and $J = 10, 15, 20, 25, 30$ (all combinations) vs. the rescaled index $(i - i_0)/N_s$ Here $i_0$ is the index of the 0-CLV and $N_s = K(1 + \alpha J)$, with $\alpha \approx 0.22$ (see text). We choose the vertical axis rescaling reference as $J_0 = 10$.

will be discussed in the next section.

Before concluding this section, we would like to briefly discuss the time-resolved projected norm $\phi_i(t)$. So far, we have discussed time-averaged quantities, but it is worth mentioning that the $X$ projections of individual CLVs are extremely inter-

5    mittent, hinting at a complex tangent-space flow structure.





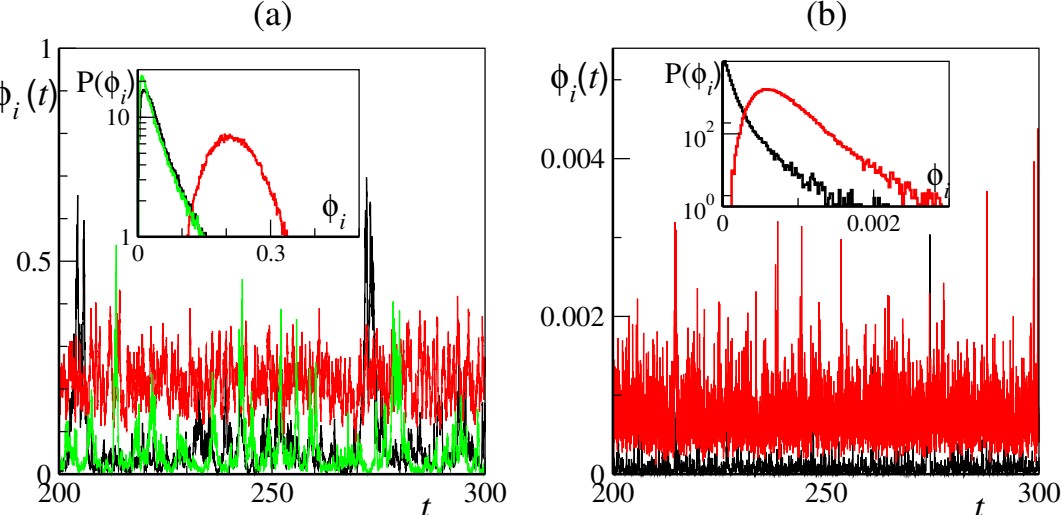

**Figure 5.** Time trace and probability distribution of CLV instantaneous projection in the $X$ subspace for $h = 1/4$, $b = 10$, and $K = 36$, $J = 10$. (a) Slow manifold vectors, $i = 110$ (black line), $i = 122$ (red line, the 0-CLV) and $i = 160$ (green line). In the inset: Corresponding probability distributions of the three CLVs time traces. (b) For two other CLVs with negligible $X$-projection, first vector (black line) and 255th vector(red line). In the inset: Corresponding probability distributions of the two CLVs time traces.

In Fig. 5a we display a few selected time series of the norm $\phi_i(t)$ for $b = 10$, $K = 36$, $J = 10$ and $h = 1/4$ (the overall picture does not change qualitatively for different choices of the coupling strength). The time series correspond to: the 110th vector, located in the left part of the central band (before the 0-CLV), the 160th vector located in the right part, and the 0-CLV (vector index $i = 122$). We clearly see a strong intermittency, resulting in a very skewed distribution of the time resolved $X$

norms. The 0-CLV is an exception, displaying more regular oscillations and a rather symmetric distribution around its mean value. This confirms the peculiar nature of the 0-CLV, whose delocalized structure is essentially determined by its alignment with the phase space flow.

In Fig. 5b we display vectors outside the slow manifold: the 1st and the 250th, which are on the left and on the right hand side of the central band, respectively. We see that also vectors outside the slow manifold display a certain degree of intermittency,

albeit on a faster time scale, and rather skewed distributions of their $\phi_i(t)$ values. Their $X$-projection, of course, is strongly suppressed and remains very close to 0. In Sec. 4, we will further comment on the intermittent behavior of $\phi_i(t)$, showing that it arises from near degeneracies in the instantaneous instability rates.

## 4   The origin of the slow manifold

In the previous section we have identified a slow manifold in the tangent space of the L96 model – a "central band" centered

around the 0-CLV – whose covariant vectors are characterized by a large projection over the slow degrees of freedom. It is





natural to expect this band to be associated not only with long time scales (i.e. the inverse of the corresponding LEs), but also with large scale-instabilities.

We begin by discussing the pedagogical example of the uncoupled system ($h = 0$). In this limit, the $X$ and $Y$ subsystems evolve, by definition, independently, and one can separately determine $K$ LEs associated with the slow variables and $KJ$ exponents associated with the fast variables. The full spectrum can be thereby reconstructed by combining the two distinct spectra into a single one. The result is illustrated in Fig. 6a, where the red crosses correspond to the $X$ LEs. Note that the same area is also spanned by the central part of the fast-variable spectrum. The region covered by the slow LEs, where the instability rates of the two uncoupled systems have the same magnitude, roughly corresponds to the location of the slow manifold in the coupled-model CLVs spectrum. This suggests that the origin of the slow manifold can be traced back to a sort of *resonance* between the slow variables and a suitable subset of the fast ones.

Note also that, in the absence of coupling, the Jacobian has a block diagonal structure, with the CLVs either belonging to the $X$, or $Y$ subspaces. The projection $\Phi_i$, therefore is strictly equal to either 0 or 1, depending on the vector type, and can be used to distinguish the two types of vectors when the full set of (uncoupled) equations is integrated simultaneously. We have verified that this is indeed the case, except for sporadic deviations (from 0 and 1) due to numerical inaccuracies, which occur when a pair of exponents belonging to the two subsystems are nearly degenerate.

We now proceed to discuss the coupled case. When the coupling is switched on, it has a double effect: (i) it modifies the overall dynamics, i.e. the evolution in phase space, Eq. (1); (ii) it affects directly the tangent-space evolution, Eq. (18), destroying the block diagonal structure of the uncoupled Jacobian. This, in turn, prevents one from identifying single LEs with either the slow or the fast dynamics. In order to be able to distinguish the two contributions also in the $h > 0$ case, we study an intermediate setup characterized by a full coupling in real space, but removing it from the tangent-space dynamics. In practice, we simulate the full nonlinear model (1), and use the resulting trajectories to "force" an *uncoupled* tangent space dynamics. This way the Jacobian matrix is still block diagonal.

Thanks to this approximation, we can define two *restricted* spectra $\lambda_k^X$ and $\lambda_j^Y$ for the slow and fast variables, respectively, and thereby recombine them into a single spectrum, by ordering the exponents from largest to the most negative one. A comparison between the resulting reconstructed spectra and the full ones (with coupling acting both in real and tangent space) shows an excellent agreement, at least in the range $h \in (0, 1]$. Two examples, for $h = 1$ and $h = 1/16$ are given in Fig. 6b. Therefore, we can conclude that the coupling in tangent space affects only marginally the values of the LEs. It is not however clear to what extent this is a general property of high-dimensional dynamics: we are not aware of similar analyses made for high-dimensional models.


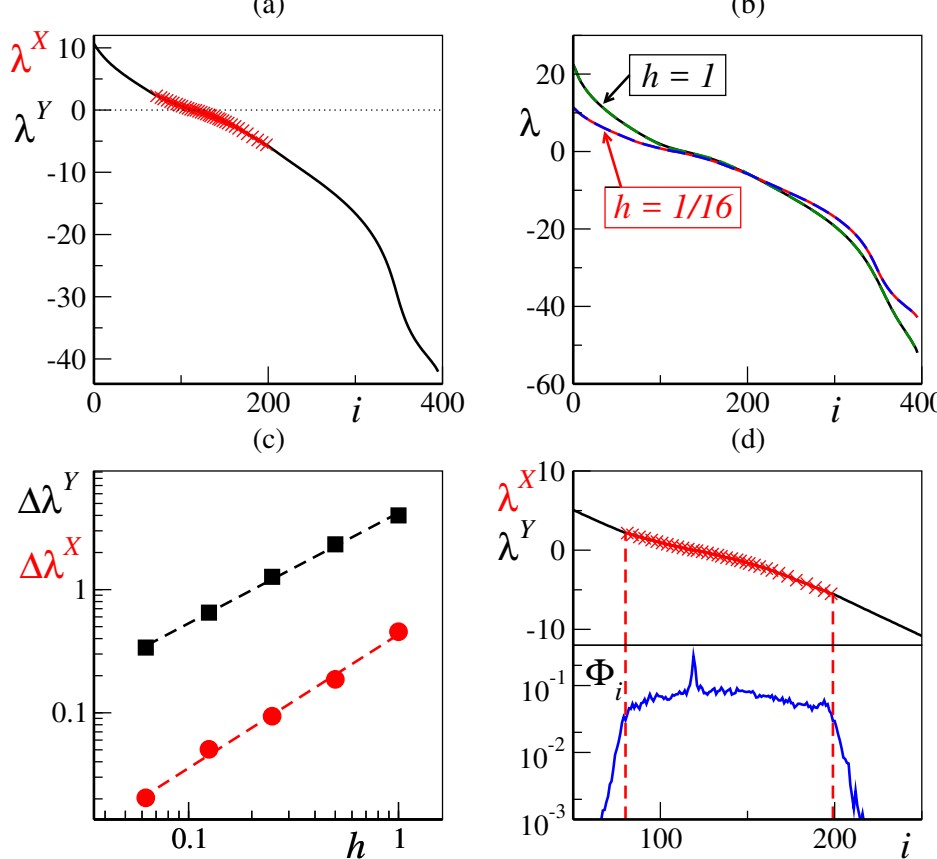

**Figure 6.** (a) Lyapunov sepctra for the uncoupled ($h = 0$) case decomposed in its slow ($\lambda^X$, red crosses) and fast ($\lambda^Y$, full black line) parts. (b) Full Lyapunov spectra (full lines) superimposed with spectra reconstructed from their restricted (see text) counterparts (dashed lines) for $h = 1$ and $h = 1/16$. (c) Root mean squared difference between the finite $h$ restricted spectra and the completely uncoupled fast and slow spectra, for both fast (black squares) and slow (red circles) dynamics. Power law fits (dashed lines) return slopes sufficiently close to unity (respectively $\approx 0.9$ and $\approx 1.1$) suggesting a simple linear behavior with the coupling $h$. Both axes are represented in a doubly-logarithmic scale. (d) Upper panel: Zoom on the central part ($i \in [50, 250]$ of the restricted spectra for $h = 1/8$ with the slow ($\lambda^X$, red crosses) and fast ($\lambda^Y$, full black line) components differently marked. Lower panel: Same zoom of the $h = 1/8$ projection norm as computed from the fully coupled dynamics. The vertical red dashed lines mark the upper ($i_L$) and lower ($i_R$) boundaries of the superposition region as reported in table 1. In all panels we have fixed $b = 10$, $K = 36$ and $J = 10$.

The modifications induced by real space coupling are more substantial. They can be quantified by computing the root-mean-square differences

$$\Delta\lambda^X(h) = \sqrt{\frac{1}{K}\sum_{k=1}^{K}\left[\lambda_k^X(h) - \lambda_k^X(0)\right]^2} \qquad (24)$$

$$\Delta\lambda^Y(h) = \sqrt{\frac{1}{KJ}\sum_{j=1}^{KJ}\left[\lambda_j^Y(h) - \lambda_j^Y(0)\right]^2},$$




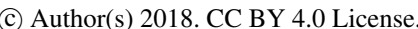

which measure the average variation of the restricted spectra upon increasing the coupling. Numerical simulations, reported in Fig. 6c, show that both $\Delta\lambda^X$ and $\Delta\lambda^Y$ increase approximately linearly with $h$, the main difference being that $\lambda_k^X(h)$ are decreased (in absolute value) when $h$ is increased, while the opposite holds for $\lambda_j^Y(h)$. While tangent-space coupling proves to be nearly irrelevant for the estimate of the LEs, average quantities computed over the entire attractor, this is not the case

of CLVs, which are local objects, defined at each attractor point. CLVs associated with the restricted LEs are, by definition, confined either to the $X$ or to the $Y$ subspace, so that $\Phi_i$ should again be either equal to 0 or 1. The analysis carried out in the previous section for the fully coupled dynamics shows instead that the average projection $\Phi_i$ of any vector within the slow manifold is significantly different from both 0 and 1 and substantially constant over the entire central band. This means that the orientation of the CLVs is very sensitive to the coupling itself.

The main mechanism responsible for the reshuffling of the CLV orientation is the (multifractal) fluctuations of finite-time LEs (Pikovsky and Politi, 2016). Fluctuations are the unavoidable consequence of the different degree of stability experienced in different regions of the phase-space and they occur in both strictly hyperbolic and non-hyperbolic dynamical systems, although they are typically much larger in the latter context. Fluctuations may be so large as to bridge the gap between distinct LEs, which results in a lack of domination of the Oseledets splitting (Pugh et al., 2004; Bochi and Viana, 2005) and in the

sporadic occurrence of near tangencies between pairs of different CLVs (Yang et al., 2009; Takeushi et al., 2011)[6]. Fluctuations are also responsible for the so-called *coupling sensitivity* (Daido, 1984; Pikovsky and Politi, 2016): strictly degenerate LEs in uncoupled systems may separate by an amount of order $1/|\ln\varepsilon|$, where $\varepsilon$ is the (small) amplitude of the coupling strength.

Let us be more quantitative and introduce the finite-time Lyapunov exponents $\gamma_i(t)$, computed from the average expansion rate over a window of length $\tau_w$,

$$\gamma_i(t) = \frac{1}{\tau_w}\ln||\mathbf{M}(\mathbf{x}_t,\tau_w)\mathbf{v}_i(t)||\,,\tag{25}$$

where $\mathbf{M}(\mathbf{x}_t,\tau_w)$ is the tangent space propagator (15) for the tangent-space evolution over time $\tau_w$, while the CLVs $\mathbf{v}_i$ is normalized to unity. Their asymptotic time average obviously coincides with the corresponding LEs, $\langle\gamma_i(t)\rangle_t \equiv \lambda_i$.

We are interested in the probability distribution $P(\gamma_i)$ of $\gamma_i$, obtained by evolving a long trajectory. In Figs. 7a-b we show the distribution for two pairs of nearby LEs within the slow manifold for $h = 1/16$ (we have set $\tau_w = 0.5$, after having verified that

it is long enough to be considered asymptotic). We have selected two vectors which, in the absence of tangent-space coupling, are of $X$- and $Y$ type, respectively. From Fig. 7a, it is clear that the amplitude of the fluctuations largely exceeds the difference between the corresponding means (i.e., the asymptotic LEs, see the vertical straight lines) and that the same holds true after restoring the coupling in tangent space (Fig. 7b).

Consistently, in Fig. 7c we show that the corresponding CLVs are characterized by non negligible near tangencies: The

probability distribution of the relative angle

$$\theta_{i,j}(t) = \arccos[\mathbf{v}_i(t)\cdot\mathbf{v}_j(t)]\,,\tag{26}$$

indeed exhibits a peak near 0.

---

[6]Perfect tangencies may occur, but only for a set of zero-measure initial conditions, such as the homoclinic tangencies in low-dimensional chaos.





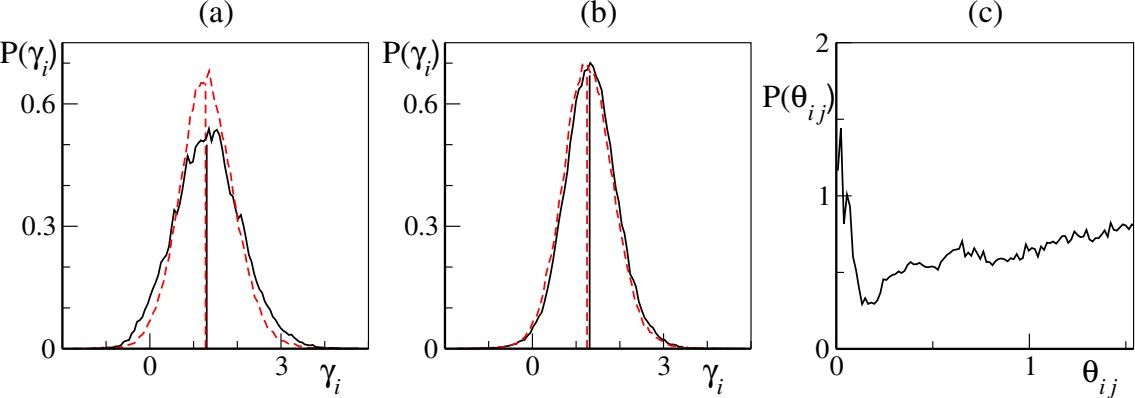

**Figure 7.** (a)-(b) Probability distribution of the finite-time LEs (25) for two nearby LEs belonging to the slow manifold. We show results for the 92nd (full black line) and 93rd (dashed red line) LEs which, in the absence of tangent space coupling (the *restricted* setup, see text), belong respectively to the $\lambda^X$ and $\lambda^Y$ restricted spectra. (a) Finite-time LEs fluctuations in the restricted case. The vertical lines mark the mean values $\lambda_{92}^X = 1.30$ and $\lambda_{93}^Y = 1.275$ (indexes refers to the full spectrum position. They correspond to the 5th and 88th (respectively) LE in the $X$ and $Y$ restricted spectra). (b) Finite-time LEs fluctuations in the fully coupled case. The vertical lines mark the mean values $\lambda_{92} = 1.31$ and $\lambda_{93} = 1.25$. (c) Probability distribution of the angle $\theta_{ij}$ (see Eq. 26) between the two corresponding CLVs in the fully coupled case. All simulations have been performed for $K = 36$, $J = 10$, $b = 10$ and $h = 1/16$. We have fixed $K = 36$ and $J = 10$ in both panels.

The near tangencies between different CLVs within the slow manifold are a proof of the "mixing" between slow and fast degrees of freedom, and perfectly consistent with the non-negligible projection on the slow $X$-subspaces of all vectors in the slow manifold. In fact, in the presence of two similarly unstable directions, the corresponding CLVs tend to wander in a (fluctuating) two-dimensional subspace selecting their current direction on the basis of the relative degree of instability. It is therefore natural to expect that, in the presence of strong fluctuations, an $X$-type vector (in the uncoupled limit) temporarily

aligns along the $Y$ directions and viceversa, thereby giving rise to a projection pattern such as the one seen in the central region of the CLVs spectrum, where all vectors have a nonnegligible average projection over the $X$ degrees of freedom.

The intermittent nature of the instantaneous $X$-projection $\phi_i(t)$ discussed in Sec. 3.1 further validates this picture: it is the result of the large fluctuations exhibited by finite-time LEs, in turn leading to fast, intermittent switching of the instability directions as the associated CLVs go through a series of near tangencies, projecting in and out of the slow degrees of freedoms.

Furthermore, the large ratio between the amplitude of the fluctuations and the separation between consecutive LEs suggests that this exchange of directions may extend beyond nearest-neighbours along the spectrum. We conjecture that the relatively smooth boundary of the central band is precisely a manifestation of this sort of long-range interaction.

Finally, we return to the restricted LEs to see whether – as implied by the above conjecture – their knowledge can help to identify the slow-manifold boundaries determined from the integration of the full tangent-space equations. More precisely,

we have identified the borders of the region covered by both slow and fast LEs. They are given by the indices (within the





reconstructed spectrum) of the largest and smallest restricted slow LEs, labelled respectively as $i_L$ and $i_R$. They are reported in Table 1, together with the corresponding value of the restricted Lyapunov exponent, for different coupling values. The agreement with the actual boundaries of the slow manifold – as revealed by a visual inspection of the projection patterns $\Phi_i$ – is actually pretty good (see, e.g., Fig. 6d for $h = 1/8$). Altogether our analysis suggests that the coupling in real space induces a

**Table 1.** $X$-restricted largest ($\lambda_1^X$) and smallest ($\lambda_K^X$) LE with the corresponding full spectrum indexes ($i_L$ and $i_R$). All data refer to $b = 10$, $K = 36$ and $J = 10$.

| $h$ | $\lambda_1^X$ | $\lambda_K^X$ | $i_L$ | $i_R$ |
|---|---|---|---|---|
| 1 | 1.33 | -4.57 | 104 | 157 |
| 1/2 | 1.88 | -5.18 | 93 | 161 |
| 1/4 | 2.095 | -5.37 | 85 | 162 |
| 1/8 | 2.16 | -5.50 | 80 | 163 |
| 1/16 | 2.25 | -5.58 | 77 | 163 |

sort of "short-range" interaction within tangent space: each LE (and the corresponding CLV) tends to affect and be affected by exponents with similar magnitude and thereby characterizing a similar degree of instability, in a sort of resonance phenomenon.

## 5   Finite perturbations

So far we have studied the geometry of the L96 model, dealing exclusively with infinitesimal perturbations. A legitimate question is whether we can learn something more, by looking at *finite* perturbations.

Finite-size analysis has been implemented in the L96 model since its introduction (Lorenz, 1996) and it has been formalized with the definition of the so-called Finite Size Lyapunov Exponents (FSLEs) (Aurell et al., 1997). In a nutshell, the rationale for introducing FSLEs is – as already recognized by Lorenz – that in nonlinear systems the response to finite perturbations may strongly depend on the observation scale. Dropping the limit of vanishing perturbations, of course, weakens the level of mathematical rigor of the infinitesimal Lyapunov analysis, but it nevertheless allows for a meaningful study of the underlying instabilities.

Here, we follow the excellent review (Cencini and Vulpiani, 2013), where applications to L96 were also discussed. Given a generic trajectory $\mathbf{x}(t)$, the idea is to define a series of thresholds $\delta_n = \delta_0 \sigma^n$, with $\sigma > 1$, and to measure the times $\tau(\delta_n)$ needed by the norm of a finite perturbation $\Delta \mathbf{x}(t) = \mathbf{x}'(t) - \mathbf{x}(t)$ to grow from the amplitude $\delta_n$ to $\delta_{n+1}$. The FSLE $\Lambda(\delta_n)$ is then defined as

$$\Lambda(\delta_n) = \frac{\ln \sigma}{\langle \tau(\delta_n) \rangle}, \tag{27}$$

where $\langle \cdot \rangle$ denotes an average over many realizations of the perturbation. In practice, one starts at time $t_0$ with a finite perturbation $\|\Delta \mathbf{x}(t_0)\| \ll \delta_0$ to ensure a correct alignment (along the most expanding direction) by the time the perturbation amplitude



reaches the first threshold $\delta_0$. Subsequentely, both trajectories $\mathbf{x}$ and $\mathbf{x}'$ are followed, registering the crossing times of all $\delta_n$ thresholds. By repeating this procedure many times, one is able to estimate the FSLEs for all amplitudes $\delta_n$ via Eq. (27). The FSLE in principle depends on the norm used to define the size of the perturbation (Cencini and Vulpiani, 2013). However, by construction, for vanishing perturbations, the FSLE should coincide with the largest LE regardless of the norm

$$\lim_{\delta \to 0} \Lambda(\delta) = \lambda_1 . \tag{28}$$

In Ref. (Cencini and Vulpiani, 2013), FSLEs have been applied to analyse the L96 model (see also (Boffetta et al., 1998) for an earlier study). In a slightly different setup (no fast-variable forcing and a larger scale separation $b$), it was shown that the FSLE is characterized by two different plateaus: (i) a small-$\delta$ one, essentially associated with the instability of the fast, convective, degrees of freedom and roughly equivalent to the largest LE, $\Lambda(\delta) \approx \lambda_1$; (ii) a "large"-$\delta$ plateau that was associated with the intrinsic instability $\Lambda_s$ on the slow larger scales. Interestingly, it was observed that also the height of the second plateau seems

to be roughly norm independent. This observation led to the conjecture that the nonlinear evolution of large perturbations may be controlled by the linear dynamics of an effective lower dimensional system, capturing the essence of the slow-variable dynamics (Cencini and Vulpiani, 2013).

In this section we repeat this analysis in our setup, comparing the behavior of the FSLE with the analysis of the tangent-space manifold. In the following we use our standard parameters ($K = 36$, $J = 10$, $b = 10$), setting $\sigma = \sqrt{2}$, $\delta_0 = 10^{-3}$ and averaging

the crossing-times over $10^3$ realizations. For each realization, the initial finite perturbation $(\Delta X_1, \ldots, \Delta X_K, \Delta Y_{1,1}, \Delta Y_{K,J})$ is chosen at random with an initial amplitude of $10^{-5}$.

As we expect the FSLE to depend on the norm, we have decided to transform this weakness into an advantage, by studying the behavior of an entire family of Euclidean norms and thereby extracting useful information from the dependence on the chosen norm. More precisely, we introduce the $\gamma$-dependent norm

$$\|\cdot\|_\gamma = \sqrt{\sum_k X_k^2 + \gamma \sum_{k,j} Y_{k,j}^2}, \tag{29}$$

which, for $\gamma = 1$, coincides with the standard Euclidean norm. We consider $\gamma \in [0,1]$, a choice which allows exploring a broad range of weights of the slow-variables. In the inset of Fig. 8a, we see that the main effect of changing the norm is a variation in the length of the two plateaus: upon decreasing $\gamma$, the first plateau shrinks, leaving space for a longer second plateau. The height of the two plateaus is largely $\gamma$-independent. This behavior can be qualitatively understood as follows. At early times, all

components of the perturbation grow according to the maximum LE, which we know from the previous analysis to be mostly controlled by the dynamics of the fast $Y$ variables. As time goes on, the perturbations of the $Y$ variables start saturating, while those of the slow variables keep growing, at a pace, however, controlled by their (weaker) intrinsic instability. Upon decreasing $\gamma$, the relative weight of the less unstable, slow variables increases. However, there is a limit: even when $\gamma = 0$, the growth rate of the $X$-perturbations is initially controlled by the fast variable. The range of this initial, fully linear, regime, depends on the

initial amplitude of the fast components: this limit corresponds to the dashed curve into the inset of Fig. 8a.

The FSLEs obtained for different coupling parameters $h$ are shown in Fig. 8a, all for $\gamma = 10^{-3}$. Two plateaus are clearly visible in all cases. The first one coincides with the maximum LE of the whole system, as per Eq. (28). The second one



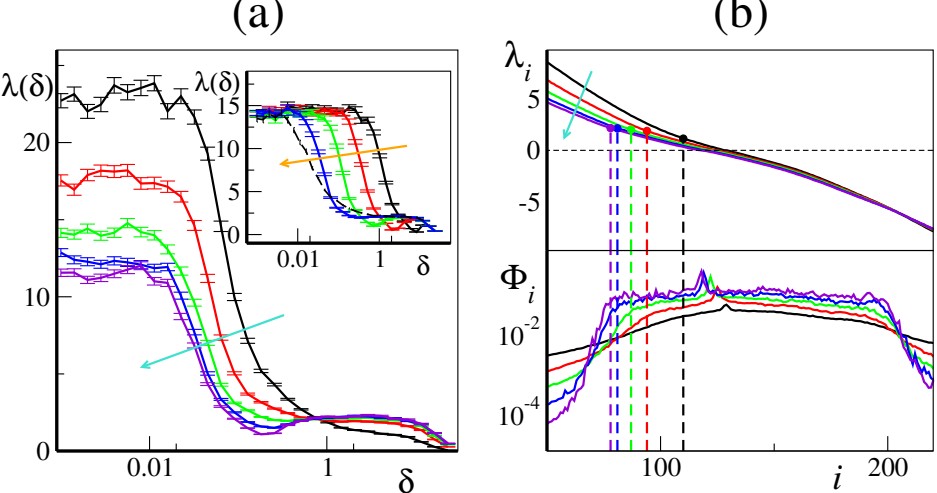

**Figure 8.** FSLE analysis. (a) FSLEs for different coupling constants, $h = 1$ (black line), $h = 1/2$ (red line), $h = 1/4$ (green line), $h = 1/8$ (blue line), $h = 1/16$ (red line) – decreasing along the cyan arrow – vs. the finite size amplitude $\delta$ for the $\gamma = 10^{-3}$ norm. Inset: FSLEs for $h = 1/4$ and different $\gamma$-norms ($\gamma = 1, 10^{-1}\, 10^{-2}\, 10^{-3}$, decreasing along the orange arrow). The error bars measure one standard error. The dashed black line marks the FSLEs for the limiting norm $\gamma = 0$. Note the logarithmic scale for the abscissa of both graphs. (b) Details of the LE spectrum (top panel) and of the $\Phi_i$ average projection norm patterns (lower panel) as in Fig. 3b. The full dots and the vertical dashed line mark the value and the location of the LEs $\lambda_{i_S}$ as identified in Table 2 (see main text for more details). Different coupling constant values are color coded as in panel (a), with $h$ decreasing along the cyan arrow in the top panel.

approximately extends over a range of one order of magnitude (at large scales the plateau is obviousy limited by the attractor size). Its height corresponds to the characteristic instability $\Lambda_s(h)$ associated with the effective dynamics of the slow variables, as conjectured in (Cencini and Vulpiani, 2013).

For each coupling $h$, we estimated the corresponding $\Lambda_s(h)$, and identified it with the closest LE, $\lambda_{i_S}$ in the Lyapunov spectra. The results of this procedure are summarized in Table 2 and compared with the results of the restricted analysis obtained in section 4.

By interpreting the height of each plateau with a suitable LE within the full spectrum, one can thereby extract the corresponding index $i_S$ for each value of the coupling constant. In Fig. 8b we have marked these labels with dashed vertical lines (upper panel) and compared with the slow manifold projection patterns (lower panel). Interestingly, these values seem to provide a convincing estimate of the leftmost boundary of the central band which defines the slow manifold in tangent space. Essentially, $i_s$ coincides fairly well with the left "shoulder" where $\Phi_i$ starts to drop towards negligible values for decreasing $i$. Note however, that for larger values of $h$, the second plateau becomes less sharply defined, up to the case $h = 1$, where it is practically impossible to define a threshold. Correspondingly, also the boundaries of the slow manifold in tangent space, as defined by inspection of $\Phi_i$, become less well defined.





**Table 2.** Estimated slow dynamics finite size instability $\Lambda_s(h)$ compared to the closest LE and its index $i_S$ for different coupling constants. $\Lambda_s$ has been typically averaged over the range $\delta \in [1, 10]$. Results of the restricted analysis carried on in section 4 are reported from table 1 for comparison. All data has been obtained with $b = 10$, $K = 36$ and $J = 10$.

| $h$ | $\Lambda_s$ | $\lambda_{i_S}$ | $i_S$ | $\lambda_1^X$ | $i_L$ |
|---|---|---|---|---|---|
| 1 | 1.2(3) | 1.15 | 110(6) | 1.33 | 104 |
| 1/2 | 1.86(11) | 1.90 | 94(2) | 1.88 | 93 |
| 1/4 | 2.07(8) | 2.04 | 87(1) | 2.095 | 85 |
| 1/8 | 2.16(11) | 2.15 | 81(1) | 2.16 | 80 |
| 1/16 | 2.2(1) | 2.18 | 78(1) | 2.25 | 77 |

Altogether, it is remarkable that from the analysis of a single pair of trajectories one can extract information that would be otherwise contained in different Lyapunov exponents; it is also interesting to notice that the nonlinear evolution of some degrees of freedom does not substantially affect the dynamics of the other degrees of freedom. Finally, the parameter $\gamma$ proves to be useful, as it helps determining the two plateaus with a better accuracy. Moreover, our analysis shows a possible relationship between the finite-size analysis of maximally unstable direction and the standard multi-direction Lyapunov analysis. In particular, the slow-variable (large-scale) instability $\Lambda_s$ emerging from the finite-size analysis roughly coincides with the upper boundary of the slow manifold LEs (i.e. the LEs associated with the CLVs having a relevant projection on the $X_k$ variables). Following the analysis of the restricted spectra carried on in the previous section, $\Lambda_s$ is also close to the first restricted LE associated with the X subspace $\lambda_1^X$, as shown in Table 2. Our result supports the earlier conjecture of Ref. (Cencini and Vulpiani, 2013) concerning the existence of an effective lower dimensional dynamics capturing the slow-variable behavior.

## 6 Conclusions

Our analysis of the tangent-space structure of the L96 model has identified a *slow manifold* within the full tangent space. It is composed of the set of covariant Lyapunov vectors characterized by a non-negligible projection over the slow degrees of freedom. Vectors in this set are associated to the smallest (in absolute value) LEs, and thus to the longest timescales. We have verified that the number of such vectors increases linearly with the total number of degrees of freedom, so that the slow-manifold dimension is an extensive quantity, and that the upper boundary of the associated LEs roughly coincides with the large-scale instability $\Lambda_s$ emerging from finite-size analysis.

The upper and lower boundaries of the slow manifold are better defined for a weak coupling $h$. However, the rescaled formulation of L96 (see Eq. (4a)) shows that the effective upward coupling (from the fast to the slow variables) is $hf = hJc/b^2$, thereby suggesting that an increase of the amplitude separation $b$ can increase the sharpness of the slow manifold boundaries even for large $h$. As reported in Fig. 9a, numerical simulations with $h = 1$ and increasing values of $b$, actually confirm this





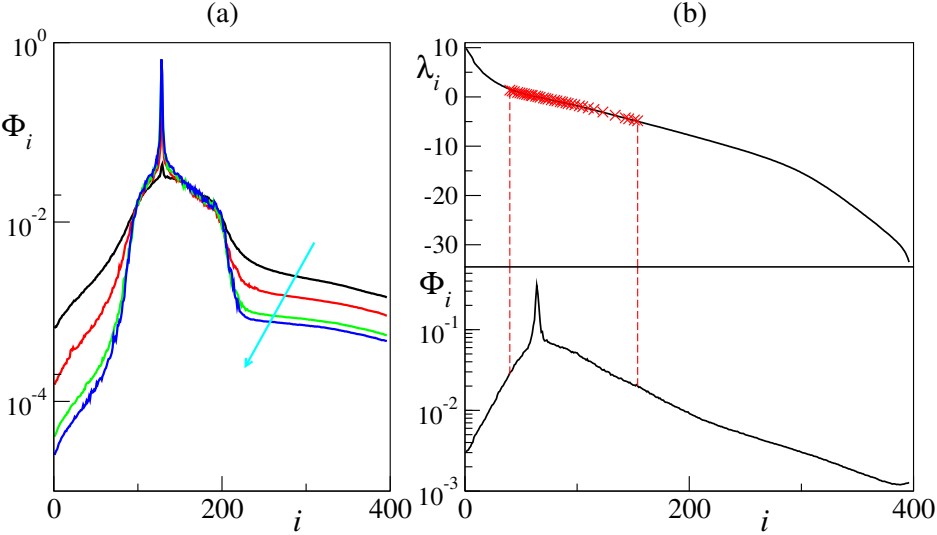

**Figure 9.** a) X-projection patterns $\Phi_i$ for Lorenz '96 with fast variable forcing ($F_f = 6$), strong coupling $h = 1$ and increasing (along the direction of the cyan arrow) values of amplitude separation, $b = 10, 20, 40, 50$. (b) Lyapunov Spectrum (top panel) and X-projection patterns (lower panel) for Lorenz '96 with no fast variable forcing ($F_f = 0$) and "standard" parameter values, $h = 1$, $b = c = 10$. The red crosses mark the values of the $X$-restricted spectrum (see Section 4 for more details). System size is $K = 36$ and $J = 10$ in both panels.

intuition, indicating that a slow manifold can be clearly defined also in the strong coupling limit, provided that the slow- and fast-scale amplitudes are sufficiently separated.

In order to clarify the origin of the slow manifold, we have introduced the notion of restricted Lyapunov spectra and argued that the central region, where the CLVs retain a significative projection over both slow and fast variables, corresponds to the range where the restricted spectra overlap with one another. In this region, fluctuations of the finite-time LEs much larger than the typical separation between consecutive LEs lead inevitably to frequent "near tangencies" between CLVs, thereby "mixing" slow fast degrees of freedom into a non-trivial set of vectors which thus carry information on both set of variables. Besides,

we have found that the coupling in tangent space hardly influences the values of LEs once the coupled dynamics is followed in real space, a property that we consider worth investigating in other high-dimensional systems.

So far, we have discussed the slow manifold in a setup where the fast degrees of freedom are forced by a strong external drive $F_f$, so that the fast dynamics is intrinsically chaotic even in the absence of coupling. One might wonder how general these results are and, in particular, how they could be extended to the traditional L96 setup, with no forcing of the fast variables

(Lorenz, 1996). When $F_f = 0$, in the zero-coupling limit, the fast dynamics is dominated by dissipation with no chaotic features. However, it is easy to verify that for a sufficiently strong coupling, the fluctuations of the slow variables induce a chaotic dynamics on the fast ones as well, so that the "classical" setup resembles the forced one analyzed in this paper. While we have not performed an accurate and thorough study, preliminary simulations indicate that a signature of a slow manifold can be found also in the classical setup for sufficiently strong coupling. In particular, as reported in Fig. 9b for $h = 1$, one can





see that the region of non-negligible $X$-projections of the CLVs again coincides with the superposition region of the slow and fast restricted spectra.

As already mentioned, the slow manifold is identified as the set of CLVs with a non negligible projection on the slow degrees of freedom. One might argue that the average projection $\Phi$ on the $X$ subspace decreases with $J$, being at best of order $1/J$, i.e. the fraction of slow degrees of freedom. However, what matters is not the actual value of $\Phi$ but rather the ratio between the height of the plateau and that of the underlying background. The scaling analysis reported in Fig. 4 shows that this ratio stays finite while increasing the number of fast variables.

Altogether, we conjecture that: (i) the fast stable directions lying beyond the slow manifold central region are basically slaved degrees of freedom, which do not contribute to the overall dynamical complexity; (ii) the fast unstable directions act as a noise generator for the $Y$ degrees of freedom (their projection on the $X$ variables being negligible, they do not *talk* directly with the slow variables). Therefore, it is natural to conjecture that the two subsystems mutually interact only through the slow manifold instabilities, so that suitably aligned perturbations of the fast variables can affect the slow variables and vice versa,

see also (Vannitsem et al , 2016). While this is a mere conjecture, to be explored in future works, it suggests that (some) covariant vectors could be profitably applied to ensemble forecasting and data assimilation in weather and climate models. In particular, it has been already shown that restricting variational data assimilation to the full unstable subspace can increase the forecasting efficiency (Trevisan and Uboldi, 2004; Trevisan et al., 2010). In the future, we would like to explore whether a data assimilation scheme restricted to the slow manifold only (which *do not* encompass the entire unstable space) can lead to further

improvements in forecasting.

In the future, it will be interesting to extend the analysis of Lyapunov exponents and covariant Lyapunov vectors to models with multiple scales and/or higher complexity and relevance, such as the coupled atmosphere-ocean model MAOOAM (De Cruz et al., 2016) or simplified multilayer models of the atmosphere such as PUMA (Fraedrich et al., 2005)) or SPEEDY (Molteni, 2003), thus going beyond the LEs studies of (De Cruz et al., 2018) to include the full tangent space geometry.

*Competing interests.* Authors have no competing interests to declare.

*Acknowledgements.* FG warmly thanks M. Cencini for truly invaluable early discussions. We acknowledge support from EU Marie Sklodowska-Curie ITN Grant No. 642563 (COSMOS). MC acknowledges financial support from the Scottish Universities Physics Alliance (SUPA), as well as Sebastian Schubert and the Meteorological institute of the University of Hamburg for the warm welcome and the stimulating discussions. VL acknowledges the support received from the DFG Sfb/Transregion TRR181 project and the EU Horizon 2020 project Blue Action.



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
