# Peer review of "Lyapunov analysis of multiscale dynamics: The slow manifold of the two-scale Lorenz '96 model"

_Nonlinear Processes in Geophysics, 2018_

## Referee Comment (RC1) · Anonymous Referee #1 · 24 Oct 2018

**1   General response and main points**

I find the results presented to be novel and interesting, but I feel that there are a number of points that warrant improvement before the manuscript can be considered for publication. To summarize my understanding of the main result:

the authors have demonstrated that in the two layer Lorenz 96 system, with respect to a scaling law in the coupling strength, the number of fast variables and relative amplitudes therein, there is a consistently identifiable, wide spectral band (close to the neutral spectrum) where the covariant Lyapunov vectors will project strongly onto the slow spatial variables. The authors conclude that there is a slow manifold structure that extends beyond the slow variables into the layer of fast variables based upon a resonance between the fast and slow variables, corresponding to similar Lyapunov spectrum when viewed in the decoupled, singular limit.

While these results are interesting and highly relevant, I feel that the conclusions and the overall analysis requires greater detail and justification.

It appears to me that there are two conceptually different notions of "slow coherent structures" that the authors wish to find a correspondence between. Firstly, there is the slow coherent structure of the $X_k$ variables which evolve on the slow time-scale in the two layer Lorenz model. On the other hand, there is the slow structure defined by the Lyapunov exponents that are close to zero, such that the corresponding covariant Lyapunov vectors exhibit weakly exponential, or sub-exponential, growth and decay. What has been established is one direction of implication: (i) we may consistently find that it is *only* a weakly unstable and stable spectral interval around zero where the associated covariant Lyapunov vectors project strongly onto the slow layer $X_k$. However, the authors have not established the other line of correspondence. Specifically, it is yet to be shown: (ii) what is the overall distribution of projection coefficients onto all spatial modes for the "close-to-neutral" spectral band. Particularly, restricting to the covariant vectors that correspond to a small threshold on the asymptotic rate of exponential growth or decay (absolute value of the Lyapunov exponent), it is of interest to find which spatial components the covariant Lyapunov vectors project most consistently onto.

An example of what would help establish the other direction of correspondence would be to show the distribution of the projection coefficients for the covariant Lyapunov vectors corresponding to a radius around the zero exponent. Particularly, it is of interest to know how strongly peaked this distribution is over the slow components $X_k$ when the spectral radius is small, as compared to when we extend the radius outwards. This would identify which spatial components, in a multiscale system, we consistently

find weakly-exponential and sub-exponential growth and decay of perturbations most frequently. The two pieces of information (i) and (ii) above would, in my opinion, give a much more complete picture together of the correspondence between these two slow structures.

A separate but important note is that the authors have not justified the use of the terminology slow "manifold". The actual manifold structure has not been sufficiently explained or supported for the understanding of the reader. To my knowledge, the existence of unstable and stable manifolds (whose tangent spaces are the sum of the unstable and stable Oseledec spaces respectively) extends to trajectories of partially hyperbolic dynamical systems, but a general construction for a center manifold doesn't extend from fixed point theory to this setting. If there is indeed a manifold structure, this point requires significant elaboration. I am satisfied with describing the above result as a "coherent structure", but the results and conclusions should be justified if using terminology with precise mathematical meaning in their statement. That said, I am in favor of the authors re-submitting when they have clarified the above points and addressed a number of minor points in the following section.

**2 Minor Points**

1. The authors frequently refer to the slow manifold in the tangent space. The global manifold structure associated to the collection of all point-wise tangent spaces (and points in the underlying base-manifold) is the tangent bundle. The meaning of the slow manifold in the tangent space should be clarified.

2. Page 4, line 17: there appears to be a typo or grammatical error in "a chaotic dynamics".

3. Page 4, line 21-22: the parameter $b$ controls the relative amplitude of the $Y$ variables, but this is unclear from the sentence.

4. Page 4, equation (2), third line: there is a typo in the sub-indices where the right hand side should have the sub-indices switched.

5. The covariant Lyapunov vectors are described as an intrinsic quantity for the system, but this is not totally accurate. The Oseledec spaces are an intrinsic quantity, but any choice of covariant vectors is only unique up to non-zero scalar when there is non-degenerate spectrum. When there is non-trivial multiplicity of exponents, there is no unique choice even up to non-zero scalars, as any choice of vectors subordinate to the Oseledec splitting can be described as covariant. This should be clarified for the reader.

6. Figure 1: this figure should be significantly re-worked as it is difficult to interpret the results.

    (a) The piece of the spectrum chosen for visualization in Fig 1.b is not spectrum of interest. The centrality of the modes closest to zero is what is of greatest significance for the analysis of slow modes, as described in Section 1. Likewise, this is important for the analysis of the numerics, as the neutral modes converge more slowly to their asymptotic limits. Particularly, the non-zero Lyapunov exponents that are close to zero become hard to distinguish from actual zero exponents numerically. It is extremely important for the analysis to understand how many zero exponents possibly exist, *even if some may be spurious* because their extremely weak exponential behavior is itself of interest.

    (b) Qualitatively, I would especially like to see how wide or narrow the "close-to-neutral" spectral band becomes when varying the parameters $K$ and $J$.

    (c) None of the circles, squares or triangles are clearly distinguishable in the figure. If we are to make any analysis based upon the correspondence between different values of $K$ and their markers, we need another, further zoomed in scale.

(d) Pesin's formula only holds for an SRB measure, while in general the sum of the positive Lyapunov exponents holds as an upper bound to the KS entropy. If there exists an SRB measure for this system, it should be clarified. If the plot for $H_{KS}$ corresponds only to an upper bound, this should be clarified.

7. Page 10, lines 3-5: the KY-dimension has only been shown equal to the information dimension of the attractor in limited cases. If it is not known that this is equal to the information dimension of the attractor, it should be clarified that the KY dimension is an approximation for the information dimension.

8. Page 10, lines 10-11: the meaning of the thermodynamic limit for the spectrum should be clarified for the reader.

9. Page 10, lines 10-15: the meaning of the statements about convergence are unclear. It appears that in the visualized part of the spectrum of Fig 1.b that there could be a limiting mode when increasing the dimension of each sector's fast layer $J$. However, in other portions of the spectrum visualized in 1.a, the evolution doesn't seem to be monotonic in $J$ (particularly in the leading and trailing exponents). Likewise, assuming that there is an asymptotic mode for the spectrum in large $J$, what is the meaning of convergence in $K$? What quantities are being compared at numerical precision? This needs clarification.

10. Figure 2.b: this plot doesn't add any new information. It would suffice to say that the Lypunov spectrum is symmetric with respect to time reversal of the dynamical system. It would again be more interesting to visualize the "close-to-neutral" spectrum in the conservative system.

11. Page 11, lines 27-28, Page 12 lines 1-9: Introducing this change of scale by $\gamma$ is confusing because the only change of scale that has been introduced thus far is with respect to the parameter $b$. This change of scale would generally change the dynamical system itself and thus the associated covariant vectors (and potentially

the spectrum). If this is not a discussion of changing the dynamical system itself, but rather simply the projection of the covariant vector into the fast variables, the meaning of the scale factor of $\gamma$ is totally unclear. As is elaborated by the authors, multiplying the fast variables by $\gamma$ and re-normalizing by the standard Euclidean distance has the effect of magnifying or de-magnifying the the projection into fast variables, and therefore *changes the span of the associated CLV*. This is mathematically unsound. If this discussion is related to the family of norms in equation (29), it should be moved into that section and the meaning should be clarified.

12. Page 12, footnote 4: the meaning of this footnote is unclear, as is where the verification of "a notable part of the energy lying in the slow variables" has taken place in the text. Please clarify this.

13. Page 12, footnote 5: the meaning of the re-indexing is unclear. Why does indexing change the convergence of the spectrum? Once again, it also needs to be clarified what is the meaning of the convergence of the spectrum within a $J$ mode as $K$ is increased.

14. Page 13, Figure 3.b: this is an interesting figure, but key information is lost based on the scale. It would be helpful for the reader to see the width of the spectral band around zero corresponding to the observed boundaries of the coherent structure in the top panel. Particularly, this is important to compare the scale of weakly-exponential and sub-exponential growth and decay that corresponds to these strong projections. It will be particularly interesting to observe, as it appears in the figure, where the boundary of the coherent structure extends beyond corresponding weakly-exponential growth and decay in the spectrum into strongly expansive or dissipative behaviour. This could be presented in an additional figure.

15. Page 16, lines 13-15: numerical verification is completely unnecessary as the

attractor, in the uncoupled limit, is decomposable into the two disjoint subsystems — any invariant ergodic measure on the attractor must also be decomposable into two ergodic invariant measures without any shared support. Oseledec's theorem, the splitting of the tangent space and the associated exponents, is stated in terms of an ergodic, invariant measure. Therefore the exponents and the splitting can be computed on each component independently.

16. Page 16, lines 19-24: this approximation of the decoupled dynamics in the fully coupled system should be made explicit. As this is a novel approximation introduced by the authors, it should be made absolutely clear what terms are being neglected in the computation of the approximately decoupled Jacobian. It would be useful for the reader to show how these terms scale with the coupling parameter. Likewise, the method of computing the Lyapunov exponents for each sub-system via this approximation should be made explicit.

17. Page 17, Figure 6.b: due to the scale in the reconstructed spectrum, a visual inspection is not informative of the similarity or dissimilarity of the actual spectrum and that which is produced via the decoupled approximation.

18. Page 18, lines 3-4: The claim that tangent-space coupling proves to be nearly irrelevant for the estimate of the LEs has not been justified quantitatively, only visually. It would be extremely helpful to study the RMSE of the distance of the true, fully coupled spectrum, from the approximate spectrum reconstructed from the decoupled approximation — this could be produced similarly to what is done in equation (24) and Fig. 6.c with the true, fully decoupled spectrum and the approximate spectrum. Claiming the approximate isolation of elements of the spectrum to either the fast or slow variables is unsupported without a quantitative measure. Particularly, the authors should explore, as a function of the coupling $h$, at what point the approximation will fail to recover an adequate reconstruction of the full spectrum. Other benchmarks of interest include: (i) how well do the

approximated exponents recover the upper bound on the KS entropy, via the sum of the positive approximated exponents versus the true ones, and (ii) how well do the approximate exponents reproduce the KY dimension computed via the true exponents.

19. Page 18, lines 25-26: it has not been explicitly specified how to associate an exponent, or the associated covariant vector, from the full spectrum with that of the approximate spectrum computed from the block Jacobian. As this is a novel approximation, it is important that this is not open to interpretation.

20. Page 19, line 1: "proof" has a precise meaning mathematically, and mathematical proof has not been evidenced in this case.

21. Page 20, lines 2-4, Figure 6.d, Table 1: This measure should again be made quantitatively. The visualization and the table are useful, but there needs to be more analysis. In particular, it would be useful to know, relative to the coupling strength: (i) what is the exact spectral interval associated to the coherent structure when studied via the exact spectrum; (ii) what is the spectral interval produced via the approximate spectrum; (ii) how many total exponents lie within each of the intervals in (i) and (ii) above; and (iv) what are the distributions of exponents in each interval studied in (iii) above.

22. Section 5: I am unable to interpret the results until the definitions and methodology are made more clear. Equation (28) has no clear definition from the quantity in equation (27) and is the primary issue for interpreting the section. What are we studying when we vary $\delta$? The FSLE defined in the equation $\Lambda(\delta_n) = \frac{\log(\sigma)}{\langle \tau(\delta_n) \rangle}$ is defined via three parameters: $n, \sigma$ and $\delta_0$. What is being varied in equation (28)? Likewise, what is $\delta$ in the horizontal axis of Figure 8.a? I will be interested in reviewing this section carefully when this is made more clear.

---

## Referee Comment (RC2) · Anonymous Referee #2 · 13 Nov 2018

**Summary**

This manuscript explores the Lyapunov spectrum of the two-scale Lorenz '96 model, composed of $K$ "slow" variables coupled to $K \times J$ "fast" variables. The main result is that the covariant Lyapunov vectors (CLVs) corresponding to Lyapunov exponents (LEs) close to zero have large average projections on the slow variables in a total energy norm. The authors claim that these CLVs constitute a tangent space "slow manifold," although the authors use this term rather loosely and do not, for example, show that this collection of CLVs is the tangent space of a proper invariant manifold. In the absence of such a demonstration, the term "slow tangent space" might be more appropriate. It seems natural that slowly growing/decaying CLVs should project onto

the slow variables, although the authors give a useful characterization the properties of this collection of slow CLVs. There are more CLVs in the slow tangent space than there are slow variables, so this tangent space necessarily includes some projection onto the fast variables which nevertheless does not induce a fast time scale. When the the slow and fast variables are decoupled, the Lyapunov spectrum of the slow variables "covers" the slow tangent space in the sense that the largest and smallest LEs of the slow variables roughly span the range of LEs of the slow tangent space. This result holds when the nonlinear systems are coupled but the tangent spaces remain uncoupled.

These results are interesting, although it's not clear whether they are particular to the L96 system or are generic to multi-scale systems. Such a determination would be a subject for further research. The present manuscript would be strengthened by a more careful characterization of the slow tangent space—in particular, whether it is actually associated with a slow manifold as the authors claim. A few of the analyses seem poorly motivated or incomplete, see major comments below. In my opinion, this paper would be publishable if these issues are addressed.

**Major Comments**

1. **Partition of the CLVs:** The CLVs in the slow tangent space have strong projections on the slow variables, but figure 5 seems to indicate a different fundamental partition of the CLVs: CLVs with a projection PDF peaked at zero (black and green curves in inset) and those whose probability of no projection onto the slow variables is *zero* (red curves in inset). I suspect that the red curves could be given a large projection onto the slow variables by a suitable rescaling of the projection norm, but the others (peaked at zero projection) could not. This is more of an observation than a critical comment, but I would be interested in the authors' thoughts on this matter.

2. **Finite-time LEs:** It is not clear what is gained by the use of the finite time LEs. They are used to show that the PDFs of the growth rates of disparate CLVs are similar. This conclusion could have been drawn from the instantaneous growth rates of the CLVs. The case for finite-time LEs is further weakened by the authors' use of a finite time window that they consider asymptotic after stating that the finite-time LEs asymptotically converge to the LEs (compare lines 22 and 24–25 on page 18). This may just be poor phasing, but it is confusing and makes the finite-time LE results difficult to interpret.

3. **Near-tangencies:** The authors show that two (out of hundreds) of the CLVs exhibit frequency near-tangencies, but for this to be a convincing explanation for the smooth edges of the slow tangent space the authors need to show that this behavior is generic; that is, that most if not all of the CLVs near the slow tangent space have frequent near-tangencies.

4. **Finite-size LEs:** The authors produce a table showing which LE corresponds to the second plateau of the finite-size LEs, but it's not clear why there should be any direct correspondence between the growth rate of a nonlinear disturbance and the LEs, which are the average growth rates of infinitesimal disturbances. They note that others have observed a similar second plateau and conjectured that this represents the growth rate of a linear instability involving a reduced set of variables. Even if this is true, its not clear why the growth rate of an instability involving a reduced set of variables should be equal to an LE from the original system. This idea needs further explanation in the context of the present paper.

The claim that the LEs corresponding to the second FSLE plateau provides a "convincing" estimate of the leftmost boundary of the "central band" of the slow tangent space (lines 9–10 on page 22) does not seem to follow from figure 8b. The vertical lines indicated in this figure are all clearly on the left side of the central band, but they don't appear to align with any obvious boundary.

**Minor Comments**

1. Note that Norwood et al. (J. Phys. A, 2013, doi:10.1088/1751-8113/46/25/254021) also studied CLVs in a multi-scale system, found "slow" CLVs that coupled onto both slow and fast variables, and noted that finite amplitude disturbances could be used to distinguish between the slow and fast systems. However, their model did not have an extensive Lyapunov spectrum, so they did not study the scaling properties of what they called "coupled" CLVs.

2. **Symmetry of the Lyapunov spectrum in the conservative case:** The authors argue that the Lyapunov spectrum is symmetric when forcing and dissipation are turned off, but the results presented don't necessarily back up this claim. Figure 2 simply plots the reflected Lyapunov spectrum on top of the original, which only shows that spectrum is symmetric to within the width of the curve. It would be more compelling to show the absolute difference between the reflected and original spectra on a semi-log scale with an appropriate estimate of the precision of the numerically computed LEs.

3. Lines 16–18 on page 1: "Additionally, usual arguments based on scale analysis, where only a limited set of scales are deemed important, usually fail because of the presence of, possibly slow, upward or downward cascade of energy and information." This sentence does not make sense; it is not clear what is trying to be said here.

4. Footnote 5: It is not clear why simply shifting the index by $1/2$ would improve the convergence of the spectrum.

5. Some minor grammatical errors:

   (a) Line 16 of page 1: "is stiff" instead of "immediately results as stiff".
   (b) Line 12 of page 2: "For some time" instead of "Since some time".

(c) Line 18 of page 2: "and" should be inserted before "(iii)".

(d) Line 29 of page 3: "main result" instead of "main results".

(e) Line 4 of page 4: "synoptic" instead of "synpoptic".

(f) Line 15 of page 5: "perhaps" instead of "pherhaps".

(g) Line 1 of page 22: "obviously" instead of "obviousy".

(h) Improper parenthesis around references: page 3, line 11; page 4, line 15; page 7, lines 28 & 29; page 21, line 5; page 22, line 3; and page 23, lines 18 & 19.

---

## Author Comment (AC1) · 19 Dec 2018

Dear Editor,

please find enclosed a revised version of the paper "Lyapunov analysis of multiscale dynamics: the slow bundle of the two-scale Lorenz '96 model" that we wish to resubmit to your attention (included here as supplement).

The remarks of the referees have been addressed in the modified version of the manuscript (and our manuscript title slightly changed according to the first referee remarks) and a detailed answer to all the points raised in the reports is provided in the other attached pdf (included as Fig.1 here).

All changes of note we have made to the manuscript have been marked in red for easy

reference. Additionally, an indexing mistake has been corrected in Table 1 and its LEs estimates have been slightly improved by newer numerical data.

Sincerely yours, The authors

Please also note the supplement to this comment: https://www.nonlin-processes-geophys-discuss.net/npg-2018-41/npg-2018-41-AC1-supplement.pdf
* * *
Dear Editor,

please find enclosed a revised version of the paper "Lyapunov analysis of multiscale dynamics: the slow bundle of the two-scale Lorenz '96 model" that we wish to resubmit to your attention.
The remarks of the referees have been addressed in the modified version of the manuscript (and our manuscript title slightly changed according to the first referee remarks) and a detailed answer to all the points raised in the reports is provided below.

All changes of note we have made to the manuscript have been marked in red for easy reference. Additionally, an indexing mistake has been corrected in Table 1 and its LEs estimates have been slightly improved by newer numerical data.

Sincerely yours,

The authors
* * *
**Reply to Referee 1**

We wish to thank the referee for carefully reading our manuscript and for judging our results novel, interesting and highly relevant.

We believe, however, that some criticisms have been induced by a misunderstanding due to our careless use of the term "slow manifold" while referring to the sub-space spanned by the effectively slow variables. For this reason, we have decided to change our terminology, and renamed the subspace "**slow bundle**".

In any case, we wish to make clear that the "wide spectral band", whose covariant Lyapunov vectors project strongly onto the slow variables, is not "close to a neutral spectrum" in an *absolute* sense. With the only exception of the single zero LE associated to the flow, the absolute value of all other LEs of the non-conservative, full L96 model is strictly larger than zero; there is no trace of any band characterized by a sub-exponential grow, which would correspond to the central manifold and make the system only partially hyperbolic.
In fact, in our numerical analysis we are able to perfectly discriminate the single zero exponent from the rest of the spectrum with a precision of one or two orders of magnitude. See for instance the example in the first figure included in this reply.

**Fig. 1.**

**Supplement:**

[revised manuscript text omitted]

---

## Author Comment (AC2) · 19 Dec 2018

Dear Editor,

please find enclosed a revised version of the paper "Lyapunov analysis of multiscale dynamics: the slow bundle of the two-scale Lorenz '96 model" that we wish to resubmit to your attention.

The remarks of the referees have been addressed in the modified version of the manuscript (and our manuscript title slightly changed according to the first referee remarks) and a detailed answer to all the points raised in the reports is provided below.

All changes of note we have made to the manuscript have been marked in red for easy reference. Additionally, an indexing mistake has been corrected in Table 1 and its LEs estimates have been slightly improved by newer numerical data.

Sincerely yours,

The authors
* * *
**Reply to Referee 1**

We wish to thank the referee for carefully reading our manuscript and for judging our results novel, interesting and highly relevant.

We believe, however, that some criticisms have been induced by a misunderstanding due to our careless use of the term "slow manifold" while referring to the sub-space spanned by the effectively slow variables. For this reason, we have decided to change our terminology, and renamed the subspace "**slow bundle**".

In any case, we wish to make clear that the "wide spectral band", whose covariant Lyapunov vectors project strongly onto the slow variables, is not "close to a neutral spectrum" in an *absolute* sense. With the only exception of the single zero LE associated to the flow, the absolute value of all other LEs of the non-conservative, full L96 model is strictly larger than zero; there is no trace of any band characterized by a sub-exponential grow, which would correspond to the central manifold and make the system only partially hyperbolic.

In fact, in our numerical analysis we are able to perfectly discriminate the single zero exponent from the rest of the spectrum with a precision of one or two orders of magnitude. See for instance the example in the first figure included in this reply.

Fig. A -- Details of the Lyapunov spectrum for h=1/2 and the parameters used in Fig. 3 of our manuscript. In the left panel we focus on the part roughly corresponding to the slow bundle. On the right, we zoom closer to the 0 LE, with the index i=125, which measured to be 0 with an accuracy of  $10^{-4}$ .

Moreover, most Lyapunov exponents (LEs) corresponding to the slow bundle are typically of order one, and – as it can be appreciated from table 1 -- the corresponding band extends roughly between +2 and -5. Therefore, these LEs are small only in the *relative* sense, that is, when compared with the larger LEs of the entire spectrum, typically one order of magnitude larger.

Furthermore -- as it can be appreciated by the example illustrated in Fig. B of this reply – all the vectors of the slow bundle but the 0 CLV are characterized by essentially the same probability distribution of the X-projection, peaked near 0 and with an exponential tail. The only exception, other than the "special" 0-CLV is represented by the two closest vectors (green curves), whose probability distribution show some sign of "hybridization" between the two shapes.

Fig. B – Probability distribution of the X-projection of the O-CLV and its closest neighbours for h=1/4 and the typical parameters used in Fig. 5 of our manuscript. LE indices and values are reported in the legend.

To summarize, we apologize for the misunderstanding, probably induced by our usage of the term "slow manifold", but we strongly remark that the tangent subspace we have identified is not related in any way to sub-exponential or weakly chaotic instabilities in the absolute sense. Its instability rates and associated timescales are those typical of the fully chaotic slow variables.

It is thus a "slow" subspace only in a relative sense, i.e. with respect to the time scales of the fast variables and of the maximum LE.

**Minor comments**

**1.** As already discussed, we agree with the referee that and we have changed accordingly our terminology.

- 2. Done
- 3. Done
- 4. Done

**5.** We have added a couple of comments, specifying that we always refer to normalized vectors and recalling the role of degeneracies.

**6. (a)** The role of Fig. 1 is not (yet) to emphasize the presence of the central band, but rather to show the way the Lyapunov spectra converge in the so-called thermodynamic limit.

We have selected the spectral region, where the convergence is slower and we still think it is the most appropriate to display these data for a generic reader.

(b) As we have clarified above, we do not claim to be in the presence of a close-to-neutral band. The central band, the way we define it, does not substantially change with K and J, as covering an instability range of a few units as, e.g. shown in Table 1.

(c) We have removed the thick lines overlapping the symbols in order to improve the readability of the figure. The inability to clearly distinguish the different symbols one from each other, however, has precisely the meaning of showing that the dependence is extremely weak. Further zooming wouldn't help much, since we are not going to discuss the convergence of the spectrum in the thermodynamic limit.

(d) We are now recalling that the Pesin formula provides only an upper bound.

**7.** Since, we have not introduced the concept of "information dimension", we don't think it makes sense to define it to claim it is bounded from above by the Kaplan-Yorke formula. We have preferred to make explicitly reference to the Kaplan-Yorke dimension as it is often done in the literature.

**8. & 9.** These two observations originate from the same problem: an inaccurate definition of the thermodynamic limit. Section 2.6 has been thoroughly rewritten. This should solve all the objections raised by the referee.

**10.** Also following the minor remark n.2 from the second referee, to better demonstrate the symmetry we have added an inset comparing the difference between the original and the reflected spectra with our numerical precision. For the reasons already discussed, we do not believe important to particularly focus our analysis on the region closer to the 0 LEs.

**11.** Here, we disagree: no any change of scale can ever change the dynamics (if the corresponding equations are properly modified). We are just facing an unavoidable problem, when different physical observables are compared. The direction of the very same vector depends on which units are being used! This is a mathematical fact and the dependence of the results on the units of measure is something that should be stated as soon as possible. For this reason, we have not postponed the analysis: we have only made some changes to the text to make the point clearer.

**12.** We have now clarified our argument.

**13.** The shift is a standard practice in the presentation of Lyapunov spectra, especially in Hamiltonian models. In fact, it ensures a perfect symmetry of the spectrum around the midpoint 1/2 of the rescaled variable range, irrespective of the value of N. We have added a proper reference to explain it. Moreover, we now discuss this point earlier (footnote 3), since the shift is first used while showing the LS collapses of Fig. 1

Following comments n. 8-9, the thermodynamic limit is now better discussed in Section 2.6.

**14.** As we already discussed, the "central region" of CLVs with a non-negligible projection on the slow variables clearly extends up to LEs magnitude of the order one or larger. This can be already seen in Table 1, but we have now added a comment to this regard when first discussing Fig. 3.

**15.** This is of course true. Our comment however, regarded the accuracy of our algorithm when computing the two uncoupled spectra at once. Anyhow, we agree that this comment is unnecessary and we have removed it.

**16.** We have now made explicit our approximation and commented that the terms ignored in tangent space are linear in the coupling parameter *h*.

**17 & 18**. We have followed the referee suggestion and we now report the root mean squared difference between the full and approximate spectra in Fig. 6c.

Global LS indicators such as the upper bound on the KS entropy or the KY dimension are reproduced with great accuracy. For instance, for h=0.5, our approximation recovers the upper bound to KS with a 0.3% accuracy and the KY dimension with even a greater accuracy.

**19.** We now detail explicitly our procedure in a footnote.

**20.** We reworded our sentence to avoid the use of the term proof.

**21.** As we now show explicitly in Fig. 6c, the original and reconstructed spectra are very close one to each other. Therefore, we do not expect variations of note in either the spectral interval or in the number of LE contained in such interval if one would decide

instead to characterize the boundary of the slow bundle using the fully coupled spectrum (and a direct visual inspection of the projection pattern). Therefore, the data report in Table 1 already gives access to the relevant information on the amplitude of the slow bundle spectral band.

Moreover, note that the Lyapunov spectrum in the slow bundle region is approximately linear. As a consequence, the corresponding LE distribution will be approximately uniform, and we do not expect it to provide any additional information.

**22.** Finite-size Lyapunov exponents are a well-known tool, and more information on them can be obtained from the excellent literature cited in section 5.

They measure the typical growth rate of a finite size perturbation of amplitude  $\delta$  in terms of the average time  $\tau$  needed to grow by a factor  $\sigma > 1$ .

The logarithm of  $\sigma$  in the definition (28) assures that the corresponding FSLE does not depend on the choice of  $\sigma$ , as long as  $\sigma$  is small enough not to bridge over qualitatively different length scales. Our choice,  $\sigma = 2^{1/2}$ , is a typical value used in the literature that has been proved to be fully adequate for the L96 model.

On the other hand, the choice of  $\delta_0$  only determines the lower bound of the amplitude range to be explored via FSLE. As we explain in the text, our algorithmic procedure measures the FSLE at the set of amplitude thresholds  $\delta_n$  for n=0,1,2,..., but FSLE can of course be measured starting from any arbitrary initial scale  $\delta$ .

When evaluated at a size  $\delta$  sufficiently small (formally in the limit  $\delta \rightarrow 0$ ), the FSLE coincides by construction with the largest LE, for any finite  $\sigma$ .

For the sake of clarity, we now label  $\delta_n$  the horizontal axis of Fig. 8.2.

All changes we have made to the manuscript have been marked in red for easy reference. Figs. A and B above have been included for the benefit of the referee, but we have not judged necessary to include them in our manuscript.
* * *
**Reply to Referee 2**

We wish to thank the second referee for his constructive remarks and for finding our results interesting.

According to the analysis of Sec. 5, the central band defining the slow bundle (in the original version: slow manifold, see below) arises due to the overlap of the Lyapunov spectra of the two sub-systems in the central part of the spectrum. Since this latter occurrence should be generic for chaotic multiscale systems, we also believe our findings to be generic and not specific to the Lorenz 96 model. This intuition will of course need to be confirmed explicitly in other multiscale systems, but we now comment on this point in the conclusions.

We provide below answers to all comments raised by the second referee. Note that following a remark of the first referee, we have decided to substitute the term "slow manifold" with the more generic "slow bundle".

**Major comments**

**1.** While the absolute value of the CLVs projection onto the slow bundle can be altered by a proper rescaling, what really defines the slow bundle is the ratio between the projection of the CLVs in the central band and the ones outside it (notice the more than two order of magnitude difference between the vertical scale in Figs. 2a-2b), which does not depend on the rescaling suggested by the referee.

Moreover, note that the red PDF in the inset of Fig. 5a belongs to the 0-CLV. As we comment in the text, this represent a unique exception due to its strongly delocalized nature.

As we discuss more extensively in our reply to the first referee, essentially all other CLVs in the slow bundle show the same projection behaviour, with an intermittent behavior and a PDF analogous to the black and green curves in the inset of Fig. 2a.

**2.** We acknowledge that our reference to the asymptotic behavior of FTLEs is confusing. In fact, the point is that FTLEs are well defined observables (i.e. independent of the set of variables used to describe the dynamics) only when the time is long enough to kill correlation so that one can construct a coordinate-independent large-deviation function. Accordingly, by "asymptotic" we mean long enough so as to ensure "universal" fluctuations. Surely, the instantaneous growth rate would not be a proper observable, as its fluctuations would be strongly coordinate-dependent. We have rephrased the entire paragraph on page 18.

**3.** We are only showing the finite-time fluctuations and angle distribution of a couple of nearby vectors for illustrative purposes, but we have indeed verified that this behavior is absolutely generic. This is to be expected in non hyperbolic systems (see the brief discussion of the existing literature before Eq. (25)), but to clarify the matter we have added a specific comment to this regard.

**4.** We first discuss our claim that the FSLE provides an estimate of the leftmost border of the slow bundle central band. On this regard, we have to respectfully disagree with the referee objection. A careful analysis of Fig. 8(b) shows that the vertical lines marking the FSLE position in the LS do align with a clear change of slope in the projection pattern  $\Phi_i$ . This beginning of a steep descent towards a negligible projection provides a good boundary for the central band. This boundary is sharper for small coupling values h, but it can be confidently identified up to at least h=1/2.

Considering the lack of a very sharp boundary for the central band, we have nevertheless slightly toned down our claim (reasonable, instead of convincing).

Coming to the first objection, our numerical analysis shows that the strongest linear instability of the slow bundle CLVs roughly coincides with the second plateau in the FSLE analysis.

Thus, infinitesimal perturbations fully contained in the slow bundle tangent subspace will grow as finite perturbations of a sufficiently large scale.

Different linear instabilities are characterized by different saturation scales, and we remark that finite but sufficiently small perturbations will initially grow according to the strongest linear instabilities they align with. Our analysis show that, as they grow in size, they pass the saturation thresholds of the fastest CLVs lying to the left of the slow bundle. After this threshold (before the final saturation) their growth is then determined by the slow bundle CLVs, i.e. the only non-saturated instabilities at this relatively large scale.

Our analysis essentially shows that the saturation scales of the faster instabilities are well separated from the ones of the slow bundle instabilities.

Moreover, we remark that our intent is not to identify the FSLE plateau with a single well defined LE, but rather with the largest expansion rate of the slow bundle subspace. We realize that our original text could have been misleading, and therefore we have slightly modified it to make our point clearer.

We are anyhow grateful to the second referee for his comment that prompted us to better clarify the implications of our finite size analysis. We have modified accordingly the final paragraph of Section 5 and the relative conclusions in Section 6.

**Minor comments**

**1.** We thank the referee for pointing out this reference. We now cite this work in our manuscript.

**2.** We agree with the referee.**

We have performed more accurate numerical simulations and checked that the symmetry is indeed verified within our numerical precision.

We have now added an inset in Fig. 2b showing the relative absolute difference between the original and reflected spectra compared with the standard deviation of our numerical estimates. An analogous inset also shows that the symmetry between the positive and negative coupling case is verified within numerical accuracy. We have modified the text accordingly.

(In Fig. 2 note that spectrum has been re-calculated because of a previous error in the determination of the simulation parameters)

**3**. What we want to point out is that simplifications where only a limited range of scales is explicitly represented, and rest completely ignored, are often inadequate. We have modified the sentence in order to clarify our argument.

**4.** The shift is a standard practice in the presentation of Lyapunov spectra, especially in Hamiltonian models. In fact, it ensures a perfect symmetry of the spectrum around the midpoint 1/2 of the rescaled variable range, irrespective of the value of N. We have added a proper reference to explain it.

Moreover, we now discuss this point earlier (footnote 3), since the shift is first used while showing the LS collapses of Fig. 1

**5.** Thank you for pointing out these misprints, we have corrected points (a)-(g). Concerning point (h), we note that our current usage of in text-citations is in agreement with the journal style guidelines:

<<In general, in-text citations can be displayed as "[...] Smith (2009) [...]", or "[...] (Smith, 2009) [...]".>>

All changes we have made to the manuscript have been marked in red for easy reference.

---

## Referee Report (RR1)

**1 General response and main points**

I first want to thank the authors for their revision, which I believe has greatly improved and clarified their work. However, while I am in favor of the work being accepted for publication, there are still some aspects that I believe should be improved, especially regarding section 5 which I was not formerly able to interpret.

I agree with the interpretation of the figures given in line 5 - line 12 (first sentence), page 22. However, there are a few issues following this. Firstly, the proposed $\gamma$-norm is not actually a norm when $\gamma = 0$; indeed, this is in fact not positive definite. This leads me to a great deal of confusion regarding lines 12 - 14 page 22, with the discussion on $\gamma = 0$. Under the proposed metric the components of the perturbation in the fast variables can be of arbitrarily large Euclidean norm and the measured size of the perturbation will still be the same. This makes the choice of perturbations completely unconstrained and I don't really understand the point this is trying to raise. I recommend removing the discussion on $\gamma = 0$ entirely.

In a separate but related point, I recommend removing the text in lines 14 - 15 page 11 and lines 1 - 9 page 12. Firstly, this discussion is distracting in section 3 as its relevance is only explored later in section 5. Moreover, I find this discussion fairly misleading in the way it describes the "orientation" of the CLV depending on the relative scales used to represent the variables in the $\gamma$-norm. The choice of the $\gamma$-norm or any other will not change the orientation of the Oseledec spaces (or their spanning vectors). In the simple case of distinct Lyapunov exponents, one can choose an arbitrary spanning set of covariant Lyapunov vectors; the choice of normalization can only define a new spanning set via a choice of non-zero scalar multiples of the original set. If it is the intention of the authors to simply discuss the relative timescale on which errors in the fast variables saturate with respect to the saturation of errors in the slow variables, this should be self contained within section 5, and discussions should focus on this point alone.

The manuscript is otherwise strong, and I think removing the two discussions above will greatly improve the work. In the following, I will outline several minor points that should also be addressed.

**2 Minor Points**

1. Page 3, lines 12 - 13: this should not start a new paragraph, include this in the previous.

2. Page 7, line 21: citation of Benettin et al. has an extra period.

3. Page 14, figure 4 caption: quotes are reversed in "central band".

4. Figure 8: remove $\gamma = 0$ case.

5. Page 22, line 14: what is fully linear? I understood the perturbation as being evolved in the nonlinear model. The difference between the control trajectory and the perturbation might evolve weakly nonlinearly, but this is again not fully linear.

6. Page 23, lines 1 - 4 and lines 14 - 15: in lines 1 - 4 it is stated that in all cases, the two plateaus are clearly visible. However, lines 14 - 15 state that when $h = 1$ it is practically impossible to find a threshold. This is contradictory and I agree with the second statement in lines 14 - 15, I didn't find a second plateau for $h = 1$ easy to detect whatsoever. The authors should explain clearly and in detail how the plateaus are chosen.

7. Page 23, lines 20 - 21: it is stated that "The parameter $\gamma$ proves to be useful in improving the accuracy of the two plateaus exhibited by the FSLEs." I don't understand why. The case where the choice of $\gamma$ is used to identify the onset of the plateau is in Figure 8.a in the inset — here $h = \frac{1}{4}$, which was already discussed as having clear plateau onsets with respect to the standard Euclidean norm. The case where a choice of $\gamma$ may be helpful in identifying a plateau would be for $h = 1$ as discussed above, but this was not performed. Currently, I don't think this statement is justified. The choice of $\gamma$ has only helped with the qualitative discussion of the saturation of errors on page 22, lines 1 - 12.

8. Page 24, lines 4 - 5: this should not start a new paragraph.

9. Page 24, lines 15 - 17, page 25 Fig. 9: a new figure and results should not be introduced in the conclusion. This figure and the discussion should be moved to section 4.

---

## Author Response (AR2)

Dear Editor,

Thank you for handling our manuscript. We are pleased to see that our reply has been found satisfactory, that our paper notably improved and the referee now only raised minor comments to our revised manuscript.

We have further modified our paper to take into account these last remarks to the extent we consider them really beneficial. In fact, since some of these comments concern more the style of the presentation than the substance of the work, in a few occasions we have preferred to keep our presentation (see the detailed reply, here below).

Our changes are highlighted in red in the revised manuscript. Finally, we have also changed the vertical labels in Fig. 8a as we spotted a misprint.

Sincerely,

The authors

**Reply to Referee 1**

*1. Presentation of the L96 model: A standard (with the exception of small-scale foreign) version L96 model using X and Y variables is presented in equation 1 and a rescaled version using X and Z variables in equation (4). The later is the version the authors appear to make use of, but most of the other appearances of the L96 model (equations 8, 18, and 24) are in terms of the X and Y variables and the original scaling. It would be clearer to pick a version and stick with it. Using the X and Z variables throughout seems like it would make the most sense.*

We agree that the switch from Y to Z to refer to the fast variables can partially puzzle a reader. However, a thorough use of the rescaled variable Z would have consequence on the definition (scale) of the fraction Φ; additionally, the variable Y is much more appropriate for an analysis of the underlying "energetics". Finally, Z has been introduced only to show how the proper structure would look like in a context where the number of variables is changed.
Therefore, we have preferred to leave the notations as they are and added a remark on page 6, to clarify our choice of notations.

*2. Table 2: It is not clear what the numbers in parentheses are. They are not described in the caption and I was unable to find an explanation in the text.*

They measure the uncertainty in the estimated value of of $\Lambda_S$ and of our estimate of $i_S$ . This is now detailed in the table caption.

**Reply to Referee 2**

We acknowledge that the referee has appreciated the efforts made to improve the quality of the presentation. We have made further adjustments to take into account the last remarks to the extent we consider possible.

In the introduction of the report, the referee criticizes our reference to the limit case $\gamma=0$ and suggests to remove the corresponding paragraph. While we agree that for $\gamma=0$ the corresponding norm is not properly defined (not being positive definite), it is also true that any $\gamma$ value strictly larger than 0 is a meaningful choice. Therefore, we have decided to carefully reword the text to avoid reference to $\gamma = 0$, mentioning instead small $\gamma$ values (as we indeed already did in the previous version in most of the cases).

As for the following remark about the orientation of the CLVs, now, instead of referring to the "orientation" we now refer to "mutual angles", which do depend on the norm adopted, therefore avoiding any possible misunderstanding.

For what concern the list of Minor points

*1. Page 3, lines 12 - 13: this should not start a new paragraph, include this in the previous.*

Done

*2. Page 7, line 21: citation of Benettin et al. has an extra period.*
*3. Page 14, Figure 4 caption: quotes are reversed in "central band".*

Both corrected.

*4. Figure 8: remove $\gamma = 0$ case.*

As discussed above, the discussion has been rephrased rather than being removed.

*5. Page 22, line 14: what is fully linear? I understood the perturbation as being evolved in the nonlinear model. The difference between the control trajectory and the perturbation might evolve weakly nonlinearly, but this is again not fully linear.*

The referee is right: "fully linear" is too strong and invalid statement. We replaced it with "approximately linear"

*6. Page 23, lines 1 - 4 and lines 14 - 15: in lines 1 - 4 it is stated that in all cases, the two plateaus are clearly visible. However, lines 14 - 15 state that when h = 1 it is practically impossible to find a threshold. This is contradictory and I agree with the second statement in lines 14 - 15, I didn't find a second plateau for h = 1 easy to detect whatsoever. The authors should explain clearly and in detail how the plateaus are chosen.*

The referee is right: for $h=1$, the presence of a second plateau is not entirely obvious. We have weakened the statement in the beginning of page 23, restricting it to $h<1$. Additionally, we have added a sentence to clarify the protocol adopted to define the height of the plateau.

*7. Page 23, lines 20 - 21: it is stated that the $\gamma$ parameter proves to be useful in improving the accuracy of the two plateaus exhibited by the FSLEs." I don't understand why. The case where the choice of $\gamma$ is used to identify the onset of the plateau is in Figure 8.a in the inset where h = 1/4 , which was already discussed as having clear plateau onsets with respect to the standard Euclidean norm. The case where a choice of may be helpful in identifying a plateau would be for h = 1 as discussed above, but this was not performed. Currently, I don't think this statement is justified. The choice of has only helped with the qualitative discussion of the saturation of errors on page 22, lines 1 - 12.*

Here we disagree with the referee: the parameter $\gamma$ is very useful, as already noted in the main text and in the caption: the curves plotted in Fig. 8 have been obtained by using a small $\gamma$ value ($10^{-3}$), since this allows for a maximal extension of the plateau. If this is still not enough to have a clear evidence for $h=1$, this does not mean that $\gamma$ is not useful to play with.

*8. Page 24, lines 4 - 5: this should not start a new paragraph.*

Done

*9. Page 24, lines 15 - 17, page 25 Fig. 9: a new figure and results should not be introduced in the conclusion. This figure and the discussion should be moved to section 4.*

Shifting Fig. 9 in section 4 would make the reading of the manuscript yet heavier. We have preferred to leave the discussion into the final section renaming it as "Discussion and conclusions" so as to make it clear its content.